# High-throughput mutagenesis identifies mutations and RNA-binding proteins controlling *CD19* splicing and CART-19 therapy resistance

Mariela Cortés-López [1,14], Laura Schulz[1,14], Mihaela Enculescu[1,14], Claudia Paret [2,3,4], Bea Spiekermann[1], Mathieu Quesnel-Vallières [5,6], Manuel Torres-Diz[7], Sebastian Unic [8], Anke Busch [1], Anna Orekhova [1], Monika Kuban[8], Mikhail Mesitov[1], Miriam M. Mulorz[1], Rawan Shraim[7,9], Fridolin Kielisch [1], Jörg Faber[2,3,4], Yoseph Barash[5], Andrei Thomas-Tikhonenko[7,10], Kathi Zarnack [11,12✉], Stefan Legewie [1,8,13✉] & Julian König [1✉]

Following CART-19 immunotherapy for B-cell acute lymphoblastic leukaemia (B-ALL), many patients relapse due to loss of the cognate CD19 epitope. Since epitope loss can be caused by aberrant *CD19* exon 2 processing, we herein investigate the regulatory code that controls *CD19* splicing. We combine high-throughput mutagenesis with mathematical modelling to quantitatively disentangle the effects of all mutations in the region comprising *CD19* exons 1-3. Thereupon, we identify ~200 single point mutations that alter *CD19* splicing and thus could predispose B-ALL patients to developing CART-19 resistance. Furthermore, we report almost 100 previously unknown splice isoforms that emerge from cryptic splice sites and likely encode non-functional CD19 proteins. We further identify *cis*-regulatory elements and *trans*-acting RNA-binding proteins that control *CD19* splicing (e.g., PTBP1 and SF3B4) and validate that loss of these factors leads to pervasive *CD19* mis-splicing. Our dataset represents a comprehensive resource for identifying predictive biomarkers for CART-19 therapy.

[1] Institute of Molecular Biology (IMB), Ackermannweg 4, 55128 Mainz, Germany. [2] Department of Pediatric Hematology/Oncology, Center for Pediatric and Adolescent Medicine, University Medical Center of the Johannes Gutenberg University Mainz, 55131 Mainz, Germany. [3] University Cancer Center (UCT), University Medical Center of the Johannes Gutenberg University Mainz, 55131 Mainz, Germany. [4] German Cancer Consortium (DKTK), site Frankfurt/ Mainz, Germany, German Cancer Research Center (DKFZ), 69120 Heidelberg, Germany. [5] Department of Genetics, Perelman School of Medicine at the University of Pennsylvania, Philadelphia, PA 19104, USA. [6] Department of Biochemistry and Biophysics, Perelman School of Medicine at the University of Pennsylvania, Philadelphia, PA 19104, USA. [7] Division of Cancer Pathobiology, Children's Hospital of Philadelphia, Philadelphia, PA 19104, USA. [8] Department of Systems Biology, Institute for Biomedical Genetics (IBMG), University of Stuttgart, Allmandring 30E, 70569 Stuttgart, Germany. [9] Department of Biomedical and Health Informatics, Children's Hospital of Philadelphia, University of Pennsylvania Perelman School of Medicine, Philadelphia, PA 19104, USA. [10] Department of Pathology & Laboratory Medicine, Perelman School of Medicine at the University of Pennsylvania, Philadelphia, PA 19104, USA. [11] Buchmann Institute for Molecular Life Sciences (BMLS), Max-von-Laue-Str. 15, 60438 Frankfurt, Germany. [12] Faculty Biological Sciences, Goethe University Frankfurt, Max-von-Laue-Str. 15, 60438 Frankfurt, Germany. [13] Stuttgart Research Center for Systems Biology (SRCSB), University of Stuttgart, Stuttgart, Germany. [14]These authors contributed equally: Mariela Cortés-López, Laura Schulz, Mihaela Enculescu. ✉email: kathi.zarnack@bmls.de; legewie@ibmg.uni-stuttgart.de; j.koenig@imb-mainz.de

B-cell acute lymphoblastic leukemia (B-ALL) is a haemato-logic malignancy that causes a significant number of childhood and adult cancer deaths. During CART-19 immunotherapy, chimeric antigen receptor-armed autologous T-cells (CARTs) are engineered to target the surface antigen CD19 on B cells by linking the single-chain variable fragment (scFv) of an anti-CD19 antibody to the intracellular signalling domain of the T-cell receptor[1]. Upon CD19 recognition, the chimeric antigen receptors activate the cytotoxic T-cells to attack the tumour cells. CART-19 therapy was recently approved for the treatment of paediatric B-ALL in the US and Europe, achieving initial remission rates up to 90%[2]. This indicates that B-ALL cells at initial screening are typically CD19-positive. Unfortunately, up to 50% of children relapse under CART-19 therapy, and response rates are even worse in adults[3,4]. Several studies reported that in 40–60% of relapse cases, the cancerous B cells become invisible to the CARTs because they lose expression of the CD19 epitope (CD19-negative)[5–8]. This recurrently involves alternative splicing of the *CD19* pre-mRNA[9–11].

Splicing involves the excision of introns and the joining of exons by the spliceosome to generate mature mRNAs. In alternative splicing, certain exons can be either included or excluded ("skipped"), resulting in different transcript isoforms. The splicing outcome at each exon is controlled by a large set of *cis*-regulatory elements in the RNA sequence which are recognised by *trans*-acting RNA-binding proteins (RBPs) that guide the spliceosome activity. It is increasingly recognised that widespread alterations in splicing are a molecular hallmark of cancer and often contribute to therapeutic resistance (reviewed in ref. [12]). For instance, intron retention, i.e., the failure to remove certain introns, often disrupts the open reading frame (ORF) with pre-mature termination codons (PTCs) and thereby compromises the expression of the encoded proteins. Consistent with the widespread splicing changes, cancer-causing driver mutations frequently occur in splice-regulatory *cis*-elements, and many splicing factors have oncogenic properties, being commonly mutated or dysregulated in cancer[12–14].

Multiple alternative splicing events in *CD19* mRNA have been described to interfere with CART-19 therapy[9,11,15–18]. Most prominently, skipping of exon 2 results in a truncated CD19 protein which is no longer presented on the cell surface and hence fails to trigger CART-19-mediated killing[9,15]. In addition, it was reported that relapsed patients showed retention of intron 2 which introduces a PTC, thereby disrupting CD19 expression[11]. Similarly, simultaneous skipping of exons 5 and 6 introduces a PTC[9]. The splicing alterations can be caused by mutations within the *CD19* gene or by changes in the expression of *trans*-acting RBPs. For instance, it has been shown that the known splicing regulator SRSF3 binds to *cis*-regulatory elements within *CD19* exon 2 to promote its inclusion[9]. Of note, alternative *CD19* isoforms showing exon 2 skipping were observed to pre-exist in patients prior to CART-19 therapy[16], suggesting that *CD19* splicing patterns may harbour predictive information and could be modulated to re-establish sensitivity to CART-19-mediated killing. However, Orlando and co-workers suggested that alternative splicing changes in B-ALL patients are present in diagnostic samples already (albeit at low frequencies) and may not contribute meaningfully to CD19 epitope loss[5]. We, therefore, set out to investigate *CD19* alternative splicing and its molecular determinants in B-ALL in more detail.

High-throughput mutagenesis screens combined with next-generation sequencing provide comprehensive insights into the regulatory code of splicing[19–22]. The interpretation of such data is challenging, as the mutation effects often depend on other mutations and are typically most pronounced at intermediate exon inclusion levels[19,20,23]. We and others have shown by mathematical modelling that kinetic models account for the context-dependence of mutation effects on splice isoforms[19,20]. By utilising these models, systems-level insights can be gained into complex *cis*-regulatory landscapes, effects of *trans*-acting RBPs, and principles of splicing regulation[19,20,24].

In this work, we combine B-ALL patient data with high-throughput mutagenesis, mathematical modelling and RBP knockdowns to comprehensively characterise *cis*-regulatory mutations and *trans*-acting RBPs controlling *CD19* exon 2 splicing. Unlike previous mutagenesis screens, we determine all intronic and exonic mutation effects in a 1.2 kb region and quantify the abundance of 100 alternative isoforms, including intron 2 retention and alternative 3′/5′ splice site usage. Many of these isoforms encode for a non-functional CD19 protein and are therefore likely to impair CART-19 therapy. By in silico analyses and RBP knockdowns, we identify *trans*-regulators of *CD19* splicing that promote the production of the therapy-impacting isoforms. Taken together, our dataset allows for a systems-level understanding of the splicing code and provides a comprehensive resource of predictive markers for CART-19 therapy resistance.

## Results

**CART-19 patients show increased *CD19* intron 2 retention after relapse.** To resolve the contribution of *CD19* splicing in CART-19 therapy, we re-analysed RNA-seq data from Orlando and co-workers[5], in which B-ALL cells of 17 patients were sequenced at initial screening and after relapse. In contrast to the original study, we expanded the analyses to intron retention events surrounding *CD19* exon 2. We found that the average frequency of intron 2 retention across patients is unexpectedly high (63%) before therapy and significantly increases to 82% after relapse (*P* value = 0.009, Wilcoxon signed-rank test; Fig. 1a, b). The trend towards higher intron 2 retention in the therapy-resistant tumours is preserved in seven out of nine individual patients that were sequenced both before therapy and after relapse (Fig. 1b). Since the resulting isoform does not encode a functional CD19 protein, this suggests that increased intron 2 retention contributes to CART-19 therapy resistance, as reported in a recent study[11].

Given the high prevalence of intron 2 retention even before CART-19 therapy, we extended our analysis to 220 B-ALL patients from the Therapeutically Applicable Research To Generate Effective Treatments (TARGET) programme. Although these patients had not been treated with CART-19, intron 2 retention appears as the predominant isoform in almost all of them (Fig. 1c, Supplementary Fig. 1a, b). This suggests that the cancer cells generally exhibit *CD19* mis-splicing. Interestingly, strong intron 2 retention is also observed in immature B-cell precursors from healthy donors, whereas it is negligible in mature B cells (Fig. 1d). Therefore, incomplete B-cell differentiation in B-ALL may be accompanied by *CD19* mis-splicing which is further aggravated during CART-19 therapy.

**Somatic mutations in relapsed patients cause splicing alterations.** To learn about the genetic causes of the splicing alterations during relapse, we took a closer look at the mutations accumulating in the B-ALL patients of the Orlando study[5]. The majority of relapsed patients (12 out of 17) harbour somatic mutations within the *CD19* gene, including frameshift insertions, deletions and single nucleotide missense variants. We selected nine mutations in exons 2 or 3 from eight patients for further analysis (Supplementary Table 1). To test for effects on splicing, we constructed a minigene reporter that harbours *CD19* exon 1–3 including the two intervening introns 1 and 2 (Fig. 1e). We confirmed that the minigene gives rise to the same transcript

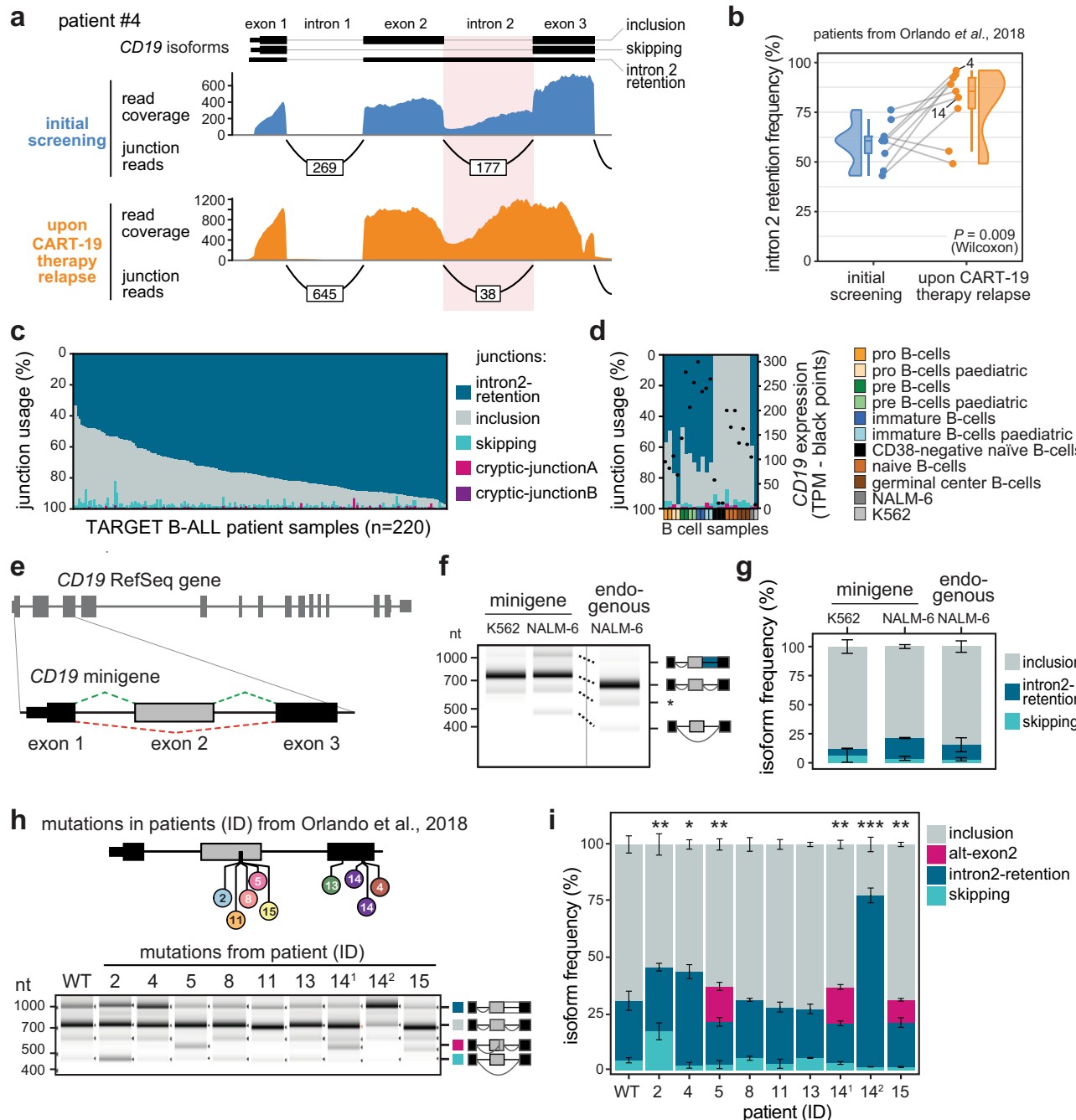

**Fig. 1 Mutations from B-ALL patients cause *CD19* mis-splicing. a** Patient #4 shows increased *CD19* intron 2 retention after CART-19 therapy relapse. Re-analysed RNA-seq data from Orlando et al.[5]. Selected isoforms (GENCODE) are shown. **b** Intron 2 retention increases in B-ALL patients after CART-19 therapy relapse. Intron 2 retention frequency (as % of all isoforms) is shown for nine patients with matched RNA-seq data at screening and after relapse. *P* value = 0.009, one-sided paired Wilcoxon signed-rank test. Grey lines connect matched samples. Boxes represent quartiles, centre lines denote 50th percentile, and whiskers extend to most extreme values within 1.5× interquartile range (IQR). **c, d** Intron2-retention is the predominant isoform in B-ALL patients and pre B cells. Stacked bar chart shows relative usage (percent selected index, PSI; left *y* axis) of junctions originating from exon 3 (Supplementary Fig. 1a) in 220 patients from the TARGET B-ALL programme (**c**) and normal B cells[44, 75] (*n* = 21) (**d**). Black dots in **d** indicate total *CD19* mRNA expression, in transcripts per million (TPM; right *y* axis). Cell lines NALM-6 and K562 are shown for comparison. **e** The *CD19* minigene spans exons 1–3 and the intervening introns from the *CD19* gene. **f, g** The minigene generates the same isoforms as the endogenous *CD19* gene in NALM-6 cells. Gel-like representation (**f**) and quantification (**g**) of semiquantitative RT-PCR showing isoforms intron2-retention (blue), inclusion (grey) and skipping (turquoise) for the WT minigene in NALM-6 cells. Isoforms of *CD19* gene in NALM-6 cells are shown for comparison. Asterisk indicates a previously reported RT-PCR artefact[66] (see Methods). Error bars indicate standard deviation of mean (s.d.m.), *n* = 3 replicates. *P* value > 0.1 for all isoforms, one-way ANOVA. **h, i** Patient mutations cause splicing changes in the *CD19* minigene. Top: Location of the tested mutations. Patient IDs as reported in Orlando et al.[5]. 14.1 and 14.2 correspond to distinct mutations from patient #14. Gel-like representation (**h**) and quantification (**i**) of semiquantitative RT-PCR as in **f, g**. Additional isoform alt-exon2 (purple) includes a truncated version of exon 2. Error bars indicate s.d.m., *n* = 3 replicates. *P* value < 0.05, **P* value < 0.01, ***P* value < 0.001, two-sided Student's *t* test. Source data including *P* values are provided as a Source Data file.

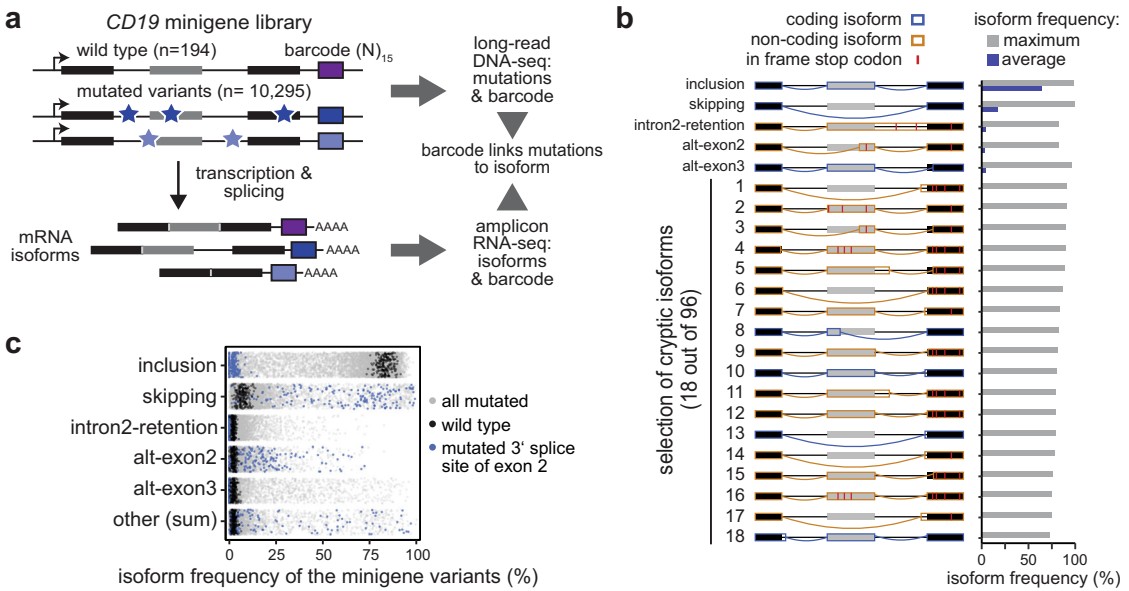

**Fig. 2 High-throughput mutagenesis identifies splicing-affecting mutations and cryptic isoforms in the *CD19* minigene. a** High-throughput detection of splicing-affecting mutations and cryptic isoforms. Mutagenic PCR creates mutated minigene variants (top) that upon transfection into NALM-6 cells give rise to alternatively spliced transcripts (bottom). Mutations (stars) and corresponding splicing products are characterised by DNA and RNA sequencing, respectively, and linked by a unique 15-nt barcode sequence in each minigene (coloured boxes). Black and grey boxes depict constitutive and alternative exons, respectively. **b** A large number of *CD19* splice isoforms arise in the minigene library. *CD19* splice isoforms with the highest maximal isoform frequency across all 9321 minigene variants. Schematic representation (left) of 5 major and 18 cryptic isoforms depicts exons 1–3 (boxes) and introns (horizontal lines) with splice junctions for each isoform (arches). Colour indicates coding potential (blue, coding; orange, non-coding). In-frame stop codons are indicated by red lines. Bar graph (right) shows average and maximal isoform frequency across all minigenes. Cryptic isoforms are sorted by maximal isoform frequency (Supplementary Data 2). **c** Inclusion isoform dominates in WT minigenes, whereas mutated variants show broad spread in all major isoforms. Frequencies of five major isoforms in replicate 1 for all wild type (black; *n* = 195) and mutated (grey; *n* = 9476) minigenes in the library. Minigene variants harbouring a mutation in the 3′ splice site of exon 2 (*n* = 174) are highlighted in blue. "Other" refers to the sum of 96 cryptic isoforms.

isoforms with quantitatively similar frequencies as the endogenous gene in the human B-ALL cell line NALM-6 (Fig. 1f, g, Supplementary Fig. 1c).

When introducing the patient mutations into our minigene reporter, we found that six out of nine tested mutations lead to the production of alternative *CD19* isoforms linked to CART-19 therapy resistance (Fig. 1h, i, Supplementary Fig. 1d): The mutation from patient #2 induces exon 2 skipping, while mutations from patients #4 and #14 (#14.2) cause intron 2 retention. The latter mirrors the increase of this isoform in the same patients after CART-19 therapy relapse (Fig. 1b). In addition, three mutations enhance the production of an additional isoform that uses an alternative 3′ splice site in exon 2 (termed alt-exon2; mutations from patients #5, #14.1 and #15). We note that as reported by Orlando and co-workers[5], most of the patient mutations also introduce frameshifts and therefore perturb CD19 protein expression by disrupting the open reading frame. Interestingly, splicing effects and frameshifts potentially interact in a non-intuitive way: For instance, the deletion in patient #5 causes a frameshift, but at the same time activates an alternative 3′ splice site (alt-exon2) which restores the open reading frame (Supplementary Fig. 1e). Thus, taking splicing information into account is essential to understand whether a targetable CD19 protein is generated in a patient harbouring *CD19* mutations.

**High-throughput screening of alternative splicing of *CD19* exons 1–3.** To systematically study the effects of point mutations on *CD19* exons 1–3 splicing, we adopted our previously developed massively parallel splicing reporter assay[19] (Fig. 2a). To this end, we randomly introduced point mutations as well as short

insertions and deletions into the *CD19* minigene reporter by error-prone PCR. This yielded a pool of 10,295 minigene variants, each with a different set of mutations and tagged with a unique 15-nt barcode sequence. As an internal control, 194 wild type (WT) minigenes with distinct barcodes were added. Mutations in all minigene variants were mapped using targeted long-read DNA sequencing (DNA-seq, PacBio SMRT-seq, Supplementary Fig. 2a, b) and validated for 30 minigene clones via Sanger sequencing. The DNA-seq data shows that the minigene variants contain on average 9.7 mutations (Supplementary Fig. 2c). This allows for a comprehensive characterisation of the mutation landscape, as each position is on average mutated in 80 different minigene variants and 90% of the mutations are present in at least four distinct minigene variants (Supplementary Fig. 2d, e). To measure splicing outcomes, the minigene pool was transfected into NALM-6 cells and the resulting transcripts were quantified by targeted RNA sequencing (RNA-seq) using 350 nt + 250 nt paired-end reads (Illumina MiSeq, Supplementary Figs. 2a, 3a). We detected around 100 different splice isoforms (see below) which were unambiguously identified by paired-end sequencing. Two replicate experiments showed a high correlation in the measured isoform frequencies (R between 0.91 and 0.98 for the different isoforms, Supplementary Fig. 3b). Based on the common barcode sequence, information from DNA and RNA sequencing could be combined, linking mutations at the DNA level to frequencies of RNA splice isoforms for a total of 10,295 minigenes in two replicate experiments (Supplementary Data 1).

**Therapy-impacting isoforms accumulate in response to numerous point mutations.** To our surprise, the screen revealed a high complexity of *CD19* exon 1–3 splicing, with a total of 101

alternative isoforms occurring with a frequency of ≥5% of all transcripts in at least two minigene variants (Supplementary Data 2). Out of these, the five major isoforms exceed 1% in WT minigenes, whereas the others, termed cryptic isoforms, only accumulate in mutated minigene variants (Fig. 2b). In WT, the most abundant major isoform by far is exon 2 inclusion (termed "inclusion"), followed by exon 2 skipping (termed "skipping") and intron 2 retention (termed "intron2-retention"). Two additional major isoforms in WT originate from alternative 3′ splice site usage within exon 2 (alt-exon2) and 3 (alt-exon3) (Fig. 2b, c). Notably, alt-exon2 uses the same splice junction that we had observed upon introducing patient mutations into the *CD19* minigene (Fig. 1i). As expected, the measured frequencies for the major isoforms show little variance for the 194 unmutated WT minigenes (standard deviation <6%, Fig. 2c). In contrast, many mutated minigene variants show strong changes relative to WT, suggesting a large impact of specific mutations on splicing outcomes (Fig. 2c). For instance, all minigenes with a mutation in the 3′ splice site of exon 2 lose the inclusion isoform, accompanied by strong alterations in the remaining major isoforms. Taken together, these observations support the accuracy of our screening results.

All major isoforms, except exon 2 inclusion, could contribute to therapy resistance either by generating an altered CD19 protein lacking a functional CART-19 epitope or by decreasing its production. Our unbiased screening approach extends the list of potentially therapy-impacting *CD19* mutations, since 1721 out of 9127 mutated minigenes show exon 2 skipping, intron 2 retention and/or alt-exon2 isoform frequencies of >25% (Fig. 2c). However, since the minigene variants carry on average 9.7 point mutations, the observed splicing changes represent the combined effects of several mutations. To extract the impact of individual mutations, we adapted our previous mathematical modelling framework[19] and implemented a multinomial logistic regression approach. Here, the splicing change in each minigene variant is described as the sum of the underlying point mutation effects (Fig. 3a, see Methods). These single-mutation effects are unknown and are determined by simultaneously fitting the model to all minigene measurements. Thereby, we were able to infer the individual effects of 4255 point mutations on the five major isoforms (Fig. 3a, Supplementary Fig. 4a). We validated the reliability of this model in describing combined mutations using a 10-fold cross-validation approach, in which we left out 10% of all minigene variants from fitting and were able to accurately predict them after model fitting (Pearson correlation coefficients 0.68–0.95; Fig. 3b, Supplementary Fig. 4b). In particular, isoforms abundant in WT or strongly accumulating in response to a large number of mutations (inclusion, skipping and alt-exon3) were predicted with high accuracy, whereas the prediction power was slightly lower for isoforms with a worse signal-to-noise ratio (intron2-retention and alt-exon2; Fig. 2c). Furthermore, we estimated that the model performed well in predicting single-mutation effects, as soon as a mutation occurred in three or more minigenes in the dataset (Supplementary Fig. 4c), which applied to 90% of all mutations (Supplementary Fig. 2e).

Out of 4255 quantified single-mutation effects, we find 193 splicing-affecting mutations that significantly alter the frequency of at least one isoform in the two replicates beyond the 2.5 and 97.5% quantiles of the WT minigene distribution (Fig. 3c, Supplementary Data 3, Source Data file). 37 of these splicing-affecting mutations overlap with single nucleotide variants (SNVs) that were previously reported in the human population from whole-genome or exome sequencing data, as well as reported cancer-associated mutations, with some of them predicted to have a pathogenic or likely pathogenic effect in the disease context (Supplementary Data 3). The strongest mutation

effects accumulate around the four main splice sites and throughout exon 2 and correspond to the core *cis*-regulatory elements, such as splice site dinucleotides, branchpoint and polypyrimidine tract, as well as auxiliary elements (Fig. 3c, d). In particular, 21% of all positions within exon 2 (55 out of 267 nt) harbour at least one splicing-affecting mutation for any isoform, suggesting that *CD19* exon 2 is densely packed with *cis*-regulatory elements.

Inspecting in more detail the 83 mutations that specifically impact *CD19* exon 2 skipping, we find them to cluster within and around exon 2 (odds ratio 2.06 for such mutations to occur inside exon 2, *P* value = 0.002614, Fisher's exact test). In addition, we observe smaller clusters of mutations within the introns and flanking constitutive exons which likely represent more distal *cis*-regulatory elements (Fig. 3c). Similarly, we explored the 54 splicing-affecting mutations impacting on intron2-retention. As expected, strongest effects are observed at the splice sites of intron 2. In addition, we find clusters of splicing-affecting mutations in intron 2 and exon 3 that might reflect important *cis*-regulatory elements. The predicted effects of all mutations on the five major isoforms can be explored in the associated Source Data file.

To test a subset of the regression predictions, we generated 19 minigenes with individual point mutations that are predicted to affect at least one isoform, including two previously reported SNVs (Supplementary Fig. 5a, Supplementary Data 4). Using semiquantitative RT-PCR, we were able to confirm that mutations near the splice sites of exon 2 predominantly lead to exon 2 skipping, whereas mutations in exon 3 result in intron2-retention and/or alt-exon3 formation (Fig. 3e, Supplementary Fig. 5b). Overall, the splicing measurements for the individual minigenes show high correlation with the regression predictions for the respective mutations across all five major isoforms (Fig. 3f, Supplementary Fig. 5c). In conclusion, our combined screening and modelling approach quantitatively describes alternative splicing of *CD19* exons 1–3 by predicting the effects of all individual point mutations and combinations thereof. Our screen thereby represents a comprehensive resource for the identification of mutations with potential clinical relevance in CART-19 therapy resistance.

**Cryptic isoforms destroy the *CD19* ORF and are associated with recurrent mutations.** Besides the five major isoforms, the *CD19* exons 1–3 can give rise to 96 cryptic isoforms which are rare (<1%) in WT, but accumulate upon certain mutations (Fig. 2b, Supplementary Data 2). The cryptic isoforms involve a total of 71 cryptic splice sites (Fig. 4a). Of note, 33 of these cryptic isoforms make up >50% of total transcripts and are therefore dominant in certain minigene variants (Fig. 2b, c). To assess whether these cryptic isoforms impact on CD19 epitope presentation, we analysed their coding potential and found that the vast majority of cryptic *CD19* isoforms (78 out of 96) show a frameshift and/or carry a PTC (Fig. 4b). This will either lead to the production of truncated CD19 peptides that likely do not allow for presentation on the cell surface[15] or will induce nonsense-mediated mRNA decay of the cryptic isoforms and will hence reduce *CD19* transcript and protein levels.

To derive a mechanistic understanding of cryptic isoform biogenesis, we analysed the underlying point mutations. To this end, we calculated a prevalence score which quantifies the degree of association between an isoform and a point mutation by multiplying: (i) the frequency of a mutation being present if the isoform level is high (>5%), and (ii) the frequency of the isoform level being high given that the mutation is present. A prevalence score of 1 indicates perfect correspondence between mutation and isoform, whereas a prevalence score of 0 is observed if they are unrelated. This score-

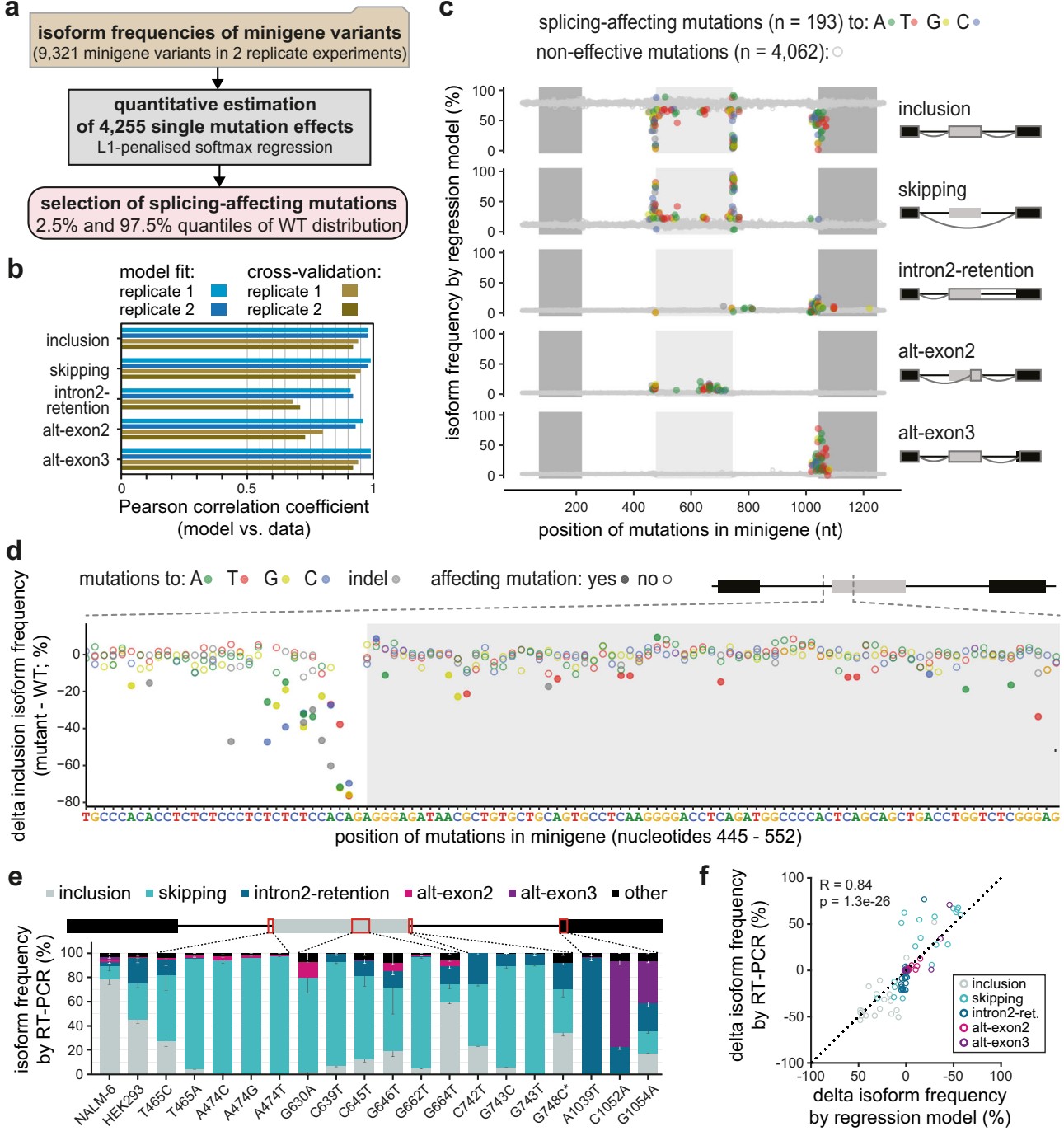

based analysis revealed 38 mutation-isoform pairs with prevalence scores > 0.25 which could explain the genesis of 36 cryptic isoforms based on 31 point mutations (Supplementary Fig. 6a, Supplementary Data 2). The remaining 60 cryptic isoforms do not show a specific association, implying that they can either be generated by multiple redundant mutations, or that our screen lacks sufficient coverage to support a reliable association. To directly test the predicted associations, we introduced five mutations with a specific association to a cryptic isoform in our minigene reporter (C535G, chr16: 28932405, prevalence score = 0.18; C806A, chr16:28932676, 0.68; A827T, chr16:28932697, 0.93; C864G, chr16:28932875, 1; G1005A, chr16:28932734, 0.89). Semiquantitative RT-PCR confirmed that all five tested mutations lead to the appearance of the associated cryptic isoform (Fig. 4c, d).

Altogether, our analysis provides a list of 31 mutations that are likely to trigger cryptic isoform formation. Importantly, the resulting cryptic isoforms show a maximum usage of up to 91% (Supplementary Data 2), which is expected to drastically interfere with normal *CD19* splicing, protein production, and subsequent epitope presentation. Screening for the occurrence of the 96 cryptic isoforms in the TARGET B-ALL patient samples, we readily detected two junctions of cryptic isoforms that had been present already prior to CART-19 therapy (Fig. 1c, Supplementary Fig. 1a). Other cryptic isoforms predicted from our screen were not found in these patients that had not been treated with CART-19 therapy, but could already exist subclonally and/or may only emerge under the selective pressures of CD19-directed immunotherapy. The same applies to the associated mutations

**Fig. 3 Quantitative modelling predicts single-mutation effects on splice isoforms. a** Based on the experimentally measured frequencies of five major isoforms in 9321 minigene variants (top box), a softmax regression model was formulated to estimate 4255 single-mutation effects (middle box) using L1 penalisation. Splicing-affecting mutations were selected for each isoform based on their respective empirical WT frequency distribution using the 2.5% and 97.5% quantiles as cutoff. **b** The model performs well in fitting and 10-fold cross-validation. Bars show Pearson correlation coefficients between model and data for two replicates and each of the five isoforms across all combined mutation minigenes considered in model training and validation, respectively (Supplementary Fig. 4a, b). **c** Splicing-affecting mutations accumulate in distinct regions around exons 2 and 3. Landscape of model-predicted single-mutation effects on five major isoforms. Predicted isoform frequencies are plotted as a function of the position of a mutation. Colours indicate nucleotide substitution of splicing-affecting point mutations (see legend), and non-effective mutations (grey). **d** Zoom-in shows model-predicted delta inclusion isoform frequency (frequency for a point mutation - frequency in WT) for nucleotides 445–552 of the minigene. Splicing-affecting mutations are highlighted as filled circles. **e** Model validation by splicing analysis of 19 minigene variants containing single point mutations. Isoform frequencies (in %) of the five major isoforms (see legend) are shown as mean values of three biological replicates (error bars, s.d.m.). 'NALM-6', splicing pattern of WT minigenes (RNA-seq) in the mutagenesis screen, 'HEK293', RT-PCR-based quantification of the baseline minigene containing mutation G742C (see Methods) in HEK293 cells. G748C* is a minigene containing G748C but lacking G742C. Schematic representation of *CD19* minigene (top) highlighting mutated regions (red rectangles). Error bar represent s.d.m., $n = 3$ replicates. **f** Splicing outcomes from **e** (y axis) are related to single-mutation predictions of the regression model (x axis; mean of two fits, each explaining one mutagenesis replicate). Changes in isoform frequency of the major isoforms (see legend) are expressed as differences (delta) relative to the baseline. Pearson correlation coefficient and *P* value (two-sided) were calculated over all isoforms (see Supplementary Fig. 5c for correlations of individual isoforms). Source data are provided as a Source Data file.

identified from our screen which were also not present in the TARGET B-ALL data (Supplementary Data 4).

**The cryptic isoforms are caused by mutations that disrupt or create splice sites.** Due to their potential clinical relevance, we wanted to learn more about how the mutations activate the cryptic isoforms. We found that the majority of mutations with a prevalence score > 0.25 are either in close proximity or directly overlap with the associated cryptic splice site (77.4% with distance <5 nt; odds ratio 7.55, *P* value = 1.793e-07, Fisher's exact test; Fig. 4e). Further inspection showed that the underlying mutations either destroy the original splice site (7.9%) or generate a new cryptic splice site (57.9%). Hence, the cryptic isoforms do originate from the generation or destruction of core *cis*-regulatory elements rather than affecting auxiliary elements.

Currently, major efforts are ongoing to implement artificial intelligence (AI) tools to predict the effect of clinical variants on the splicing outcome. We therefore tested whether the state-of-the-art neural network SpliceAI[25], which predicts changes in the splicing patterns induced by single point mutations, captures the gain and loss of splice sites in *CD19*. We applied SpliceAI using all possible single-point mutations in the *CD19* minigene as an input. Similar to the results from our mutagenesis screen (Fig. 4a), SpliceAI predicts cryptic splice site activation by mutations throughout the minigene, with an increased density around the 3′ splice site of exon 3 (Supplementary Fig. 6b). All SpliceAI-predicted mutations are close to the affected cryptic splice sites (Supplementary Fig. 6c). Hence, SpliceAI successfully reflects the global landscape of mutation-induced cryptic splice site activation in the *CD19* minigene.

With respect to the accuracy of the individual predictions, we found that 10 out of 38 mutations with strong SpliceAI predictions (SpliceAI score > 0.5) indeed lead to the accumulation of splice isoforms with the corresponding cryptic splice sites in the experimental data (prevalence score > 0.25, Fig. 4f). In the remaining 28 cases, either weak overall cryptic splice site activation occurred in the data (9 cases) or a different cryptic splice site was activated than predicted by SpliceAI (19 cases; Supplementary Fig. 6b). In quantitative terms, the likelihood of a cryptic splice site activation according to the SpliceAI prediction ("SpliceAI score") is correlated to the magnitude of the prevalence score linking the mutation to the corresponding cryptic isoform in our screen (Fig. 4f). Overall, the comparison supports that SpliceAI can guide the interpretation of mutation effects in clinical samples, though direct experimental validation is necessary. Due to the robust performance of SpliceAI, we decided

to predict splice-changing mutations throughout the entire *CD19* gene and overlapped them with publicly reported single nucleotide variants (SNVs; Supplementary Fig. 6c). These predictions and variant overlap are provided as a resource (Supplementary Data 5) and can be used to evaluate the impact of new patient mutations on *CD19* splicing in the future.

From our mutagenesis data, we found that the cryptic isoforms arise from numerous 3′ and 5′ cryptic splice sites that distribute over the entire minigene and accumulate at exon 3 (Fig. 4a). In line with their high prevalence, 26 cryptic splice sites reach >50% usage upon certain mutations, particularly around the start of exon 3. We hypothesised that cryptic splice site activation occurs in exon 3 because its canonical splice site can be outcompeted by neighbouring cryptic sites. To test this, we scored the strength of local consensus sequences using MaxEntScan[26], and indeed found that the 3′ splice site of exon 3 is weak compared to all other canonical splice sites of *CD19* exons 1–3 (Fig. 4g, Supplementary Fig. 6e, f). In line with our hypothesis, mutations around the 3′ splice site of exon 3 frequently create stronger splice sites than elsewhere in the minigene that exceed the strength of the canonical 3′ splice site of exon 3 (Fig. 4g). This suggests that weak splice sites are particularly vulnerable to the activation of competing cryptic splice sites and should be of particular interest when assessing the impact of clinical variants on splicing outcomes.

**An extensive network of RBP regulators might drive *CD19* mis-splicing.** Besides *CD19* mutations, CART-19 therapy resistance may also stem from altered expression of *trans*-acting RBPs which bind to the *CD19* pre-mRNA to control alternative splicing. To identify putative RBP regulators, we explored publicly available databases containing experimentally determined RBP binding motifs (ATtRACT[27] and oRNAment[28]). Furthermore, we included RBP binding information from the public resource of ENCODE eCLIP datasets[29]. Since the *CD19* mRNA is hardly expressed in the ENCODE cell lines and binding events in *CD19* can therefore not be directly extracted, we employed the prediction algorithm DeepRiPe[30]. The underlying neural network has been trained on the PAR-CLIP and ENCODE eCLIP datasets and thereby allows to predict changes of RBP binding upon mutation in any RNA sequence. In combination, these tools predict a total of 198 RBPs to bind within *CD19* exons 1–3 (ATtRACT: 62 RBPs; oRNAment: 70 RBPs) or to decrease (or increase) binding upon mutation (DeepRiPe: 128 RBPs; Fig. 5a, b, Supplementary Fig. 7a).

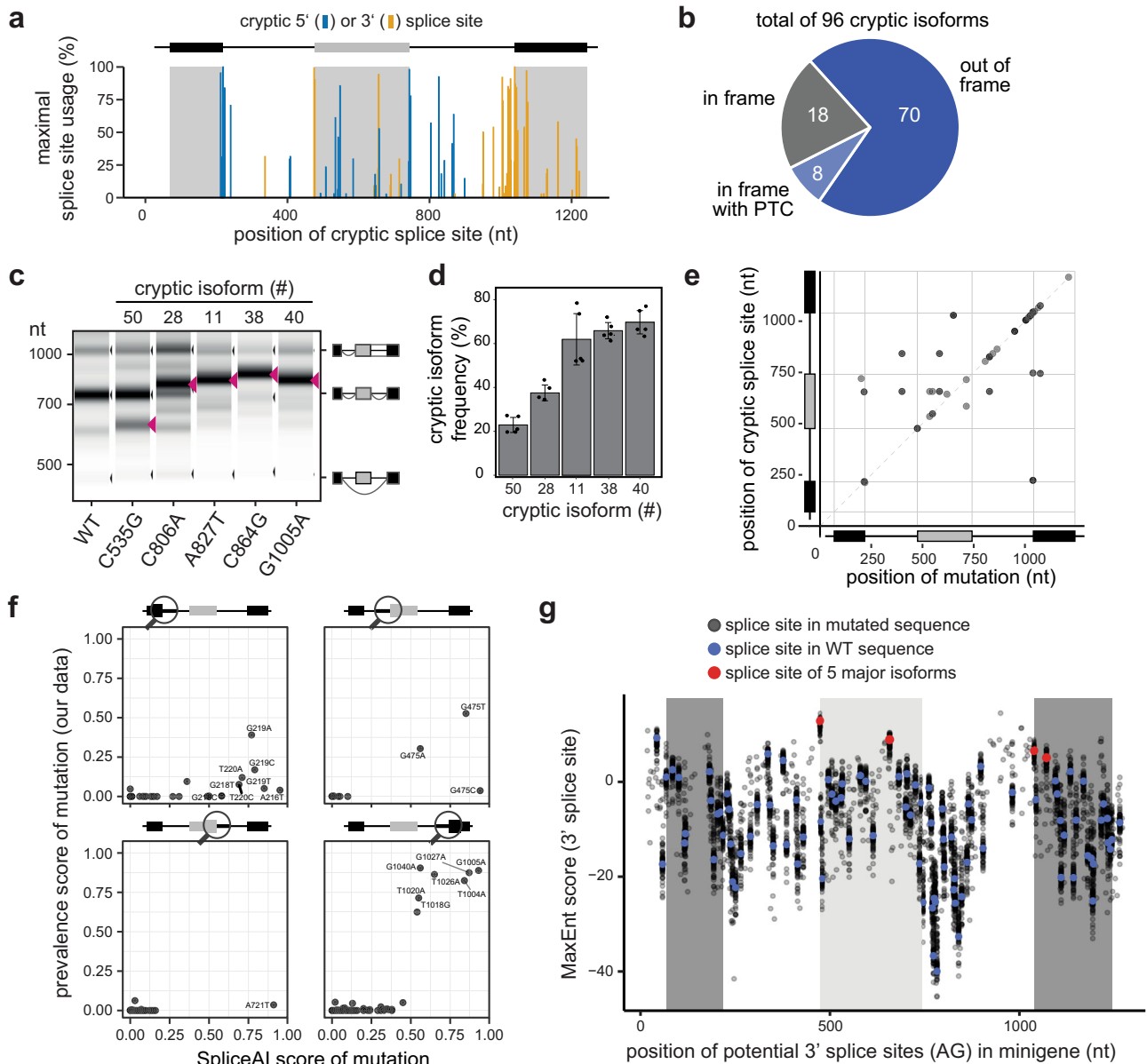

**Fig. 4 *CD19* mutations frequently activate cryptic splice sites. a** Alternative splicing of *CD19* minigene variants involves 71 cryptic splice sites. Splice site usage (sum of junction-spanning reads in a splice site / total number of reads in minigene) was calculated for each minigene variant. Maximum usage across all minigenes is plotted against the corresponding position to the cryptic splice sites. **b** Cryptic isoforms code for non-functional CD19 proteins. Out of 96 cryptic isoforms, 8 run into a premature termination codon (PTC) and 70 are out-of-frame. The remaining 18 remain in frame, but are shortened or extended relative to the reference inclusion isoform. **c**, **d** Experimental validation of five point mutations associated with distinct cryptic isoforms. Predicted cryptic isoforms are indicated by red arrowheads. Gel-like representation (**c**), with major isoforms indicated on the right, and RT-PCR quantification (**d**). Error bars indicate s.d.m., $n = 3$ replicates. **e** Mutations leading to cryptic isoforms are often located within or near cryptic splice sites. For 31 cryptic isoforms that are highly associated with a mutation (prevalence score > 0.25; y axis), the position of the mutation (x axis) was related to the used cryptic splice site (y axis). **f** SpliceAI correctly predicts single mutations leading to the generation of cryptic isoforms. SpliceAI scores of 0–1 reflecting the probability to gain a cryptic splice site in response to a mutation (see Methods). Scatterplots compare the SpliceAI score against the prevalence score (association of a mutation with a cryptic isoform) from our data, for 254 mutation-splice site pairs that match in their positions with SpliceAI. Separate panels are shown for each canonical splice site (circle in schematic minigene representation). **g** Exon 3 harbours a weak 3′ splice site and is preceded by many potentially competing cryptic 3′ splice sites. Dotplot shows splice site strengths (MaxEnt score) for putative 3′ splice sites (AG dinucleotides) in the *CD19* minigenes. MaxEnt score was calculated in 23-nt sliding window for WT sequence (red and blue dots) and hypothetical mutant minigenes with all possible single-point mutations (grey dots). 3′ splice sites used in the five major isoforms are highlighted in red. Source data are provided as a Source Data file.

To link the putative RBP regulators to the observed splicing changes, we overlaid the predicted binding sites (or predicted mutations for DeepRiPe) with splicing-affecting mutations from our screen. Overall, we find that 79% and 60% of ATtRACT and oRNAment binding sites, respectively, overlap with a splicing-affecting mutation (affecting any of the five major isoforms). Furthermore, 105 (5%) of the mutations predicted to change RBP binding by DeepRiPe overlap with splicing-affecting mutations, suggesting that modulating RBP binding at these sites may have a functional impact on *CD19* splicing (Fig. 5c, Supplementary

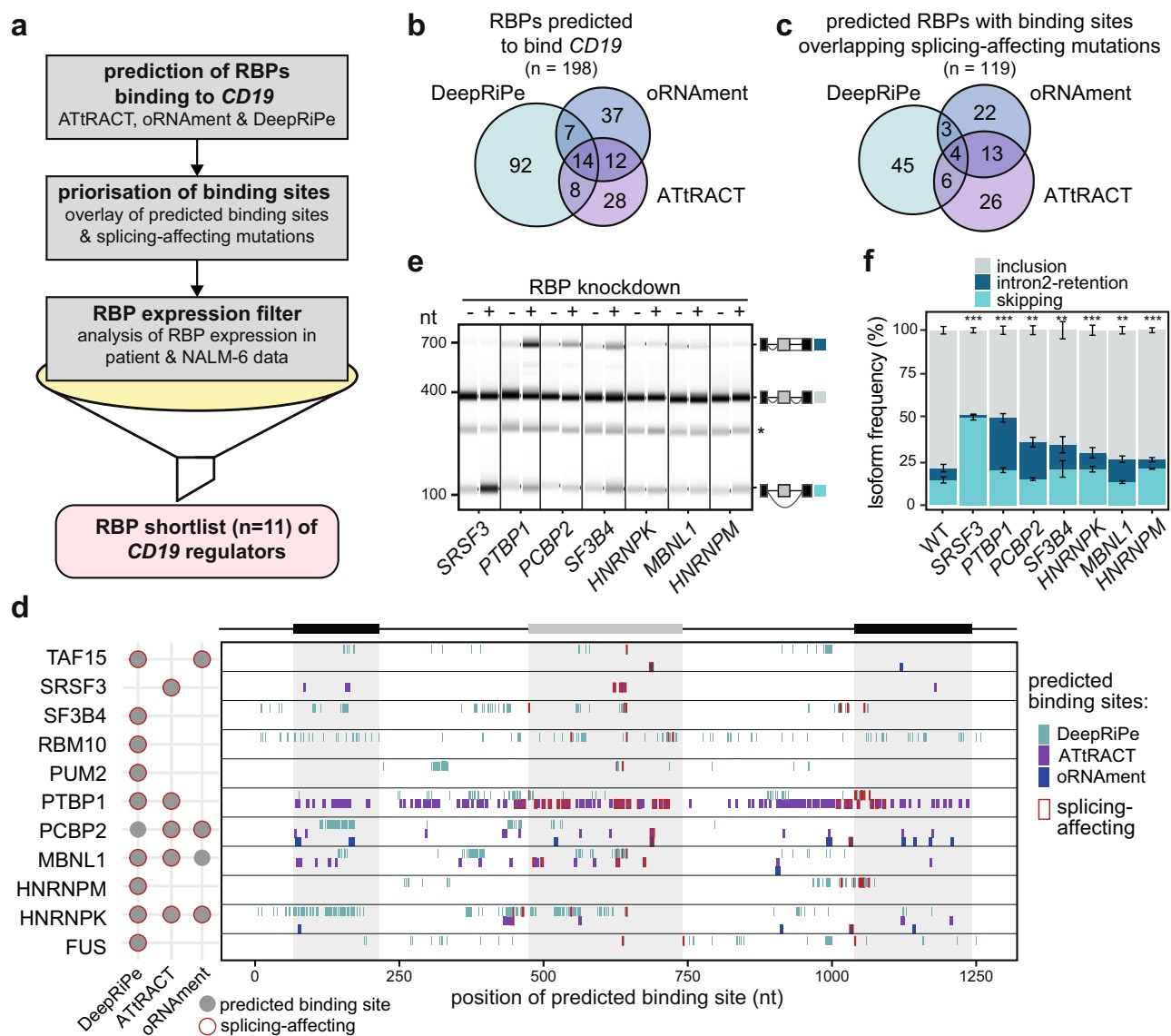

**Fig. 5 In silico predictions identify RBP regulators of *CD19* alternative splicing. a** Pipeline for the identification of potential RBP regulators of *CD19* splicing. Starting with in silico predictions, we obtained 198 candidate RBPs with predicted binding motifs (ATtRACT/oRNAment) or predicted differential binding upon mutation (DeepRiPe). These were prioritised by overlapping with the splicing-affecting mutations from our screen. Additionally, based on publicly available RNA-seq data, we required a minimum mean expression in RNA-seq data from B-ALL patients[31] and NALM-6 cells[75]. Together with literature information, we shortlisted 11 candidate RBPs for knockdown (KD) experiments, including SRSF3 as a positive control. **b, c** In silico analyses predict dozens of RBPs binding to *CD19*. Venn diagrams show overlap of RBPs in initial predictions (**b**) and after overlay with splicing-affecting mutations (**c**). **d** The 11 candidate RBPs are predicted to bind throughout the *CD19* minigene region. For each RBP, the binding sites predicted by ATtRACT and oRNAment and disrupting mutations predicted by DeepRiPe are indicated (see legend). Sites overlapping with splicing-affecting mutations are framed in red. The schematic summary (left) shows that all 11 candidate RBPs have at least one predicted site that overlaps with a splicing-affecting mutation. A full list of predicted binding sites (ATtRACT/oRNAment) and differential binding mutations (DeepRiPe) is provided in Supplementary Data 6. **e, f** Seven RBP KDs significantly change *CD19* splicing. Gel-like representation (**e**) and quantification (**f**) of semiquantitative RT-PCR showing detected isoforms exon 2 inclusion (grey), intron 2 retention (blue), and skipping (turquoise) from the endogenous *CD19* gene in KD and control NALM-6 cells. Asterisk indicates a previously reported RT-PCR artefact[66] (see methods). Error bars indicate s.d.m., $n = 3$ replicates. **$P$ value < 0.01, ***$P$ value < 0.001, two-sided Student's $t$ test. Measurements for all 11 KD experiments are shown in Supplementary Fig. 8c, d. Source data including $P$ values are provided as a Source Data file.

Fig. 7a). By merging these sets, we obtained a list of 119 RBPs that may regulate splicing by binding to *CD19* exons 1–3 (Supplementary Data 6). Most of these are expressed in cancerous B cells from B-ALL patients from[31] (80 with mean FPKM [fragments per kilobase of transcript per million mapped reads] > 10; Supplementary Fig. 7b) and could thus modulate *CD19* splicing. Among these RBPs are SRSF3, a previously reported regulator of *CD19* splicing[9], but also new candidates such as PTBP1. Overall, the in silico predictions suggest the presence of an extensive RBP

network that controls *CD19* splicing and may impact the CART-19 therapy success.

**Depletion of PTBP1 and several other RBPs results in non-functional *CD19* isoforms.** Based on our experimental data, in silico predictions, expression, literature information and manual curation, we shortlisted 11 RBP candidates for further analysis, including SRSF3 as a positive control (Fig. 5d). All 11 RBPs are

expressed in normal B cells and B-ALL patient samples from the TARGET B-ALL cohort (Supplementary Fig. 8a). To test their impact on endogenous *CD19* splicing, we generated NALM-6 cell lines stably expressing shRNAs against the shortlisted RBPs (depletion to <40% transcripts; Supplementary Fig. 8b). As previously described[9], knockdown of *SRSF3* leads to increased exon 2 skipping in the endogenous *CD19* transcripts, confirming that this SR protein is required for exon 2 inclusion (Fig. 5e, f). Importantly, we find that knockdown of six additional RBPs (*PTBP1, PCBP2, SF3B4, HNRNPK, MBNL1* and *HNRNPM)* has significant effects on *CD19* alternative splicing (Fig. 5e, f, Supplementary Fig. 8c, d). The knockdown of these factors reduces *CD19* exon 2 inclusion, while promoting intron2-retention and/or exon 2 skipping, thus shifting the cells towards expression of the relapse-associated *CD19* isoforms. This implies that reduced levels of these factors can impair targetable CD19 epitope expression.

PTBP1 stands out among the putative regulators as it shows the strongest effects on intron2-retention. This splicing event introduces a premature termination codon that likely reduces *CD19* transcript and protein expression via nonsense-mediated mRNA decay (Fig. 2b). In line with a role of PTBP1 in *CD19* missplicing in tumours, we find that patient samples from the TARGET B-ALL cohort on average show lower *PTBP1* mRNA expression compared to healthy B cells (Supplementary Fig. 8a). Within the B-ALL samples, *PTBP1* expression negatively correlates with *CD19* intron2-retention, as expected based on our knockdown experiments ($R = -0.24$; Fig. 6a, left). In addition, we investigated *PTBP2* mRNA expression, which is tightly repressed by the PTBP1 protein via alternative splicing and nonsense-mediated mRNA decay[32] and hence serves as a direct sensor for PTBP1 activity in the cells. Indeed, we find a strong correlation between increased *PTBP2* mRNA levels, i.e., lowered PTBP1 protein activity, and increased *CD19* intron2-retention ($R = 0.56$; Fig. 6a, right). To test for changes upon CART-19 relapse, we extracted *PTBP1* and *PTBP2* from expression data provided by the Orlando study[5]. Although we do not detect systematic changes in the *PTBP1* mRNAs levels, the *PTBP2* mRNA levels are significantly increased at relapse relative to screening, possibly indicating lowered PTBP1 protein levels (P value = 0.037, Wilcoxon rank-sum test; Fig. 6b). Together, these analyses suggest that PTBP1 is a regulator of *CD19* alternative splicing, which we decided to explore further.

PTBP1 recognises clusters of UC-rich motifs[33,34]. Remarkably, ATtRACT predicts almost 100 such PTBP1 binding motifs across the studied *CD19* region, including 25 that overlap with splicing-affecting mutations (Fig. 5d, Supplementary Data 6). Moreover, DeepRiPe predicts 78 mutations in 63 positions that change PTBP1 binding, out of which 10 are splicing-affecting in our screen (odds ratio 3.21, P value = 0.002481, Fisher's exact test). The high number of predicted binding sites suggests a partial redundancy, indicating that PTBP1 regulation might be difficult to disrupt with individual point mutations as introduced in our screen. To experimentally test if PTBP1 binds to the predicted sites, we performed PTBP1 iCLIP2 experiments[35] in NALM-6 cells. In line with a role in intron2-retention, we find extensive PTBP1 binding, particularly in intron 2, where it spreads over an extended cluster of predicted binding sites (Fig. 6c). This suggests that PTBP1 directly regulates *CD19* splicing via intron 2 binding.

Next, we chose to assess whether PTBP1-mediated splicing changes affect CD19 surface exposure on B cells. To test this, we performed siRNA-mediated knockdown of *PTBP1* in P493-6 and MHHCALL4 cells (Supplementary Fig. 9a, b) and confirmed that the knockdown increased levels of *CD19* intron2-retention in both cell lines (Supplementary Fig. 9c, d). Then, we measured CD19 protein surface expression using CD19 antibody staining

and flow cytometry analysis (Supplementary Fig. 9e). Strikingly, we found that both cell lines show reduced CD19 surface exposure upon *PTBP1* depletion (Fig. 6d–f, Supplementary Fig. 9f). Thus, by interfering with CD19 protein expression on the cell surface, *PTBP1* depletion could indeed contribute to CART-19 therapy resistance.

Taken together, our data suggest that both *cis*-acting mutations and *trans*-acting RBPs can lead to unproductive *CD19* splicing which in turn disrupts CD19 epitope presentation. Therefore, the splicing-affecting mutations and RBP regulators identified in this work may harbour predictive information for CART-19 therapy success.

## Discussion

Massively parallel reporter assays such as our high-throughput mutagenesis screen provide comprehensive insights into the regulatory code of splicing, as they characterise the complete set of *cis*-acting sequence mutations and reveal the binding sites of *trans*-acting RNA-binding proteins (e.g.,[19–21,36–38]). The interpretation of these datasets is challenging due to nonlinear interactions of individual mutation effects. For instance, competition effects in splicing reduce the impact of individual mutations at low and high isoform frequencies, i.e., depending on the mutational background[19,20]. In addition, other factors such as RBP expression patterns and cell type/tissue identity determine the effects of sequence mutations. Using kinetic modelling, we and others derived regression models taking competition in splicing into account, thereby showing that the effects of complex mutation combinations can be quantitatively described as the sum of individual mutation effects[19,20]. Thus, mutations seem to control splicing additively rather than synergistically, and this principle also holds for *CD19* splicing.

In our *CD19* mutagenesis dataset, we comprehensively characterise the full set of splice isoforms generated in response to thousands of sequence mutations. In particular, we find that cryptic splice site activation and thus alternative 3′ and 5′ splice site usage are common modes of alternative splicing. Intriguingly, such events do not require extensive sequence remodelling, but can often be triggered by single point mutations, as indicated by strong associations between putative cryptic isoforms and certain nucleotide substitutions. This suggests, in accordance with previous reports[39], that neighbouring splice sites frequently compete for spliceosome assembly, especially if the canonical splice site is comparably weak. While this finding shows the enormous isoform complexity that can arise already from such a simple exon configuration, it raises the question of how protein function can be robustly maintained, since most cryptic *CD19* splicing isoforms likely encode non-functional proteins.

Unlike previous mutagenesis screens which mainly focused on exonic sequence mutations, the present *CD19* dataset characterises the complete set of intronic and exonic mutations in a 1200 nt sequence stretch. The complete characterisation of *CD19* exons 1–3 required the use of long-read sequencing technology. Given that introns in human protein-coding genes on average span ~8.1 kb (GENCODE v31), the long-read sequencing methodology described in this work opens the approach for broad applications. For *CD19*, we find that strong mutation effects are mainly centred around canonical and cryptic splice sites, whereas in other examples such as *MST1R* exon 11 or *FAS* exon 6, mutation effects are more dispersed across intronic and exonic sequences[19,40]. This suggests that *CD19* exon 2 splicing may be controlled by multiple splicing enhancers that act redundantly and render inclusion less sensitive to individual point mutations[20]. Therefore, *CD19* exon 2 may require more specific perturbations and as we show here, does not only respond with

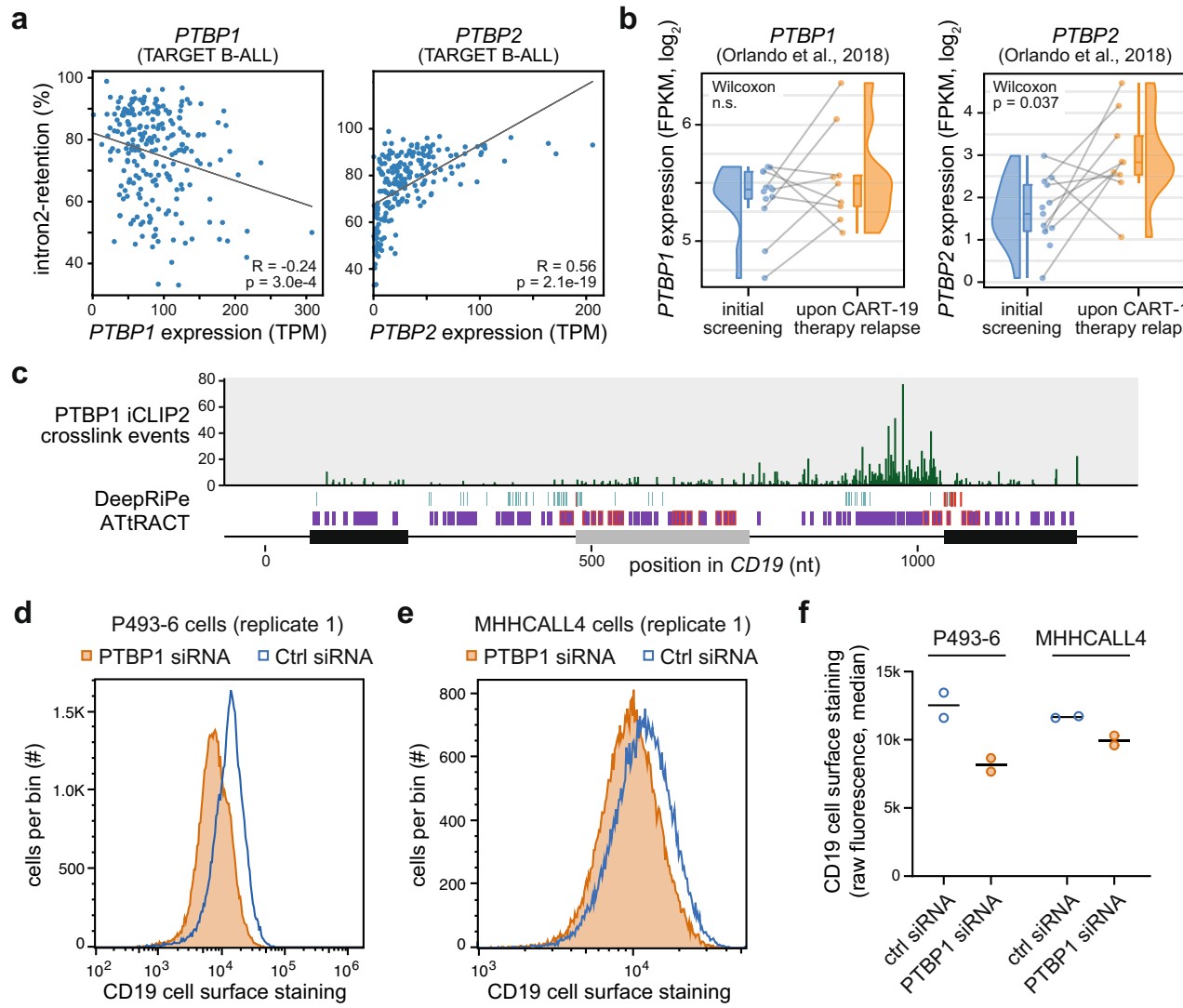

**Fig. 6 PTBP1 is a regulator of *CD19* alternative splicing. a** *PTBP1* and *PTBP2* mRNA levels correlate with *CD19* intron2-retention. Scatterplots comparing mRNA levels (TPM) to intron2-retention frequency for 220 B-ALL patient samples from TARGET B-ALL data. Pearson correlation coefficients and *P* values (two-sided) are given. **b** *PTBP2* mRNA levels are increased upon CART-19 therapy relapse. Box and violin plots showing *PTBP1* and *PTBP2* mRNA expression (fragments per kilobase of transcript per million mapped reads, FPKM) for patient samples (*n* = 9) from initial screening and upon CART-19 therapy relapse[5]. Grey lines connect matched samples of the same patients before therapy and after relapse. Boxes represent quartiles, centre lines denote 50th percentile, and whiskers extend to most extreme values within 1.5× IQR. One-sided paired Wilcoxon signed-rank test. **c** PTBP1 shows extensive binding to *CD19* intron 2. Bar diagram shows the number of PTBP1 iCLIP2 crosslink events from NALM-6 cells on each nucleotide in endogenous *CD19* exons 1–3. Predicted PTBP1 binding motifs (ATtRACT) and mutations predicted to alter PTBP1 binding (DeepRiPe) are shown below (see legend in Fig. 5d). Nucleotide positions are given relative to minigene sequence. **d–f** CD19 cell surface staining is reduced upon *PTBP1* knockdown in P493-6 (**d**) and MHHCALL4 (**e**) cells. Distributions of CD19 surface protein, as measured in 45–50 × 10³ cells (per replicate) by CD19 antibody staining and flow cytometry, in cells transfected with *PTBP1* siRNA (orange) or non-targeting control siRNA (blue). **f** Dotplot shows mean and data points for measurements of cell surface CD19 in replicate 1 (**d**, **e**) and 2 (Supplementary Fig. 9f).

exon skipping, but tends to employ alternative splice sites and intron retention, both of which are clinically relevant in the case of CART-19 therapy resistance.

Our retrospective analyses of clinical B-ALL samples implicate unproductive *CD19* splice isoforms in the development of CART-19 therapy resistance. Using minigene assays, we directly show that *CD19* mutations that are observed in relapsed patients lead to exon 2 skipping, intron 2 retention or an additional isoform that uses an alternative 3′ splice site in exon 2. Furthermore, based on our mutational scan, we report ~200 additional point mutations that significantly affect these and other therapy-impacting isoforms. Thus, our results indicate that far more *CD19* mutations can create isoforms that would escape CART-19 recognition. In

the future, targeted CRISPR/Cas9 replacement experiments using the endogenous *CD19* gene should be performed to validate that the predicted mutations cause physiological changes in splicing, loss of CD19 protein exposure on cell surface and CART-19 therapy resistance. Furthermore, the detection of such mutations in longitudinal samples may provide predictive biomarkers for therapy response in the future.

Likewise, alterations in the expression of *trans*-acting RBPs can induce aberrant *CD19* splicing, explaining the emergence of CD19-negative relapses in samples without mutations or with low-allelic-frequency mutations or without mutations in the *CD19* locus. Interestingly, we find that the differentiation status of B cells affects *CD19* splicing: in mature B cells, almost complete

exon 2 inclusion occurs, implying that all *CD19* transcripts give rise to functional CD19 protein. In contrast, intron 2 retention occurs in approximately half of the *CD19* transcripts in undifferentiated B-cell precursors (Fig. 1d). Likewise, retention of intron 2 is predominant in B-ALL patient samples from the TARGET B-ALL cohort, with 93% of patients exhibiting retention frequencies above 50% (Supplementary Fig. 1b). Hence, incomplete B-cell differentiation in B-ALL may induce a transcriptional and posttranscriptional programme, likely involving altered RBP expression, that reduces (but does not completely eliminate) the functional CD19 protein pool. This partial intron 2 retention may predispose the cancer cells to therapy resistance before they are actually subjected to CART-19 treatment, as observed in sorted B-cell populations from a B-ALL patients before and after CART-19 therapy relapse[17]. For the development of complete CART-19 resistance, some B-ALL patients thus likely host subclonal CD19-negative B-ALL cells which are further selected under the treatment[17]. The causes of complete CD19 loss in these subclonal cell populations are likely to be manifold, involving (epi)genetic changes such as hypermethylation of the *CD19* promoter[41], mutations in the *CD19* gene, splicing factor expression and combinations thereof.

Mutations in splicing factors such as SRSF2, SF3B1 and U2AF1 are common in myelodysplastic syndrome/acute myelogenous leukemia[42] and chronic lymphocytic leukemia[43], and are associated with aberrant splicing. In B-ALL, mutations in splicing factors are not common, but previous work suggests that splicing factor expression is deregulated[44]. In the context of *CD19*, we confirm that SRSF3 deregulation induces exon 2 skipping[9] and identify several other RBPs that promote the expression CD19 protein isoforms invisible to the immunotherapeutic agent, including PTBP1, PCBP2, SF3B4, HNRNPK, MBNL1 and HNRNPM. Several of the newly identified regulators have been found as deregulated in other cancer types and are discussed as potential targets for anti-cancer therapy[45–47]. In addition, upregulation of PTBP1 has been implicated in acquired resistance to the chemotherapeutic agent gemcitabine in pancreatic ductal carcinoma cells[48]. In the context of lymphocytes, PTBP1 is upregulated in B cells and required for early B-cell selection[49]. It was reported, however, that treatment of leukemic cells with the targeted therapy drug imatinib, which inactivates the BCR-ABL kinase encoded by the translocated Philadelphia (Ph) chromosome, lowers PTBP1 levels[50]. In the light of our finding that *PTBP1* knockdown increases *CD19* intron 2 retention and thereby reduces CD19 epitope presentation, previous treatments with imatinib may have negative impact on the subsequent response to the CART-19 therapy in a subset of Ph+ B-ALL patients. In addition, a recent study showed that the repeat RNA *PNCTR* sequesters substantial amounts of nuclear PTBP1 in various cancers[51]. Thus, in addition to regulation of PTBP1 expression, other factors such as availability may also influence PTBP1-mediated regulation in B-ALL cells under CART-19 therapy.

Currently, we cannot predict which patients with a CD19-positive B-ALL have a high risk of developing CD19-negative relapses. The pre-existence of isoforms skipping exon 2 or exons 5–6 has been previously discussed as a possible biomarker[16,17]. Moreover, in a comparison of B cells from a B-ALL patient, it was found that intron 2 retention had already occurred prior to CART-19 therapy (CD19-positive B cells) and had become predominant in the CD19-negative B cells after relapse[17]. Our results point to the need to extend the analysis to additional *CD19* isoforms and to incorporate the expression of splicing factors in screening approaches to identify patients at risk of relapse on CART-19 therapy. Notably, the same biomarkers might also be relevant for other malignancies arising from B-cell lineage, such as large B-cell lymphoma. Loss of CD19 following CART-19 therapy has been described as a mechanism for relapse[52], accounting for 60% of relapses in recent clinical studies[53]. Our data show that *CD19* splicing is highly complex, with already ~100 alternative isoforms concerning just exons 1–3. Of them, ~80% encode for a CD19 protein lacking a functional CART-19 epitope and are thus expected to contribute to therapy resistance. The specific detection of alternative splicing might serve as a reliable biomarker and may provide a novel approach to monitor disease progression as already suggested in other tumour entities[54]. To assess the role of the predicted cryptic splice isoforms in patients, we screened sequencing data from the TARGET B-ALL cohort and indeed recurrently found two junctions from the cryptic isoforms that we had observed in the mutagenesis data. Even though other cryptic junctions were absent and mutations associated with cryptic isoforms according to screen were also not found in the patient data, these may still emerge during CART-19 selection. Currently, there is a shortage of large-scale sequencing data of patient material before and after CART-19 therapy[55]. Future analysis of such data with a special focus on cryptic splice site usage will be important to identify mutations or splice isoforms that are predictive for CART-19 therapy success.

The contribution of aberrant splicing to CART-19 resistance may further be relevant for future combination therapies. Small-molecule splicing modulators are currently in clinical trials for myeloid neoplasms and splice-switching antisense oligonucleotides are in development for different targets (reviewed in[12]). Our mutagenesis dataset provides a strong basis for designing and systematically evaluating splice-switching oligonucleotides for the modulation of *CD19* splicing. The combined application of these splicing modulators with immunotherapy may represent a way to limit the generation of resistance to CART therapies.

## Methods

**Cell lines.** NALM-6 cells were obtained from ATCC and cultured in RPMI medium (Life Technologies) with 10% fetal bovine serum (Life Technologies) and 1% L-glutamine (Life Technologies). HEK293T cells were obtained from DSMZ and grown with the same additives as for NALM-6. For validation experiments (Fig. 3e), HEK293 cells were obtained from DSMZ and were cultured in Gibco Dulbecco's Modified Eagle Medium (DMEM, Thermo Fisher Scientific) with L-Glutamine + 10% Gibco foetal bovine serum (FBS, Thermo Fisher Scientific). All cells were kept at 37 °C in a humidified incubator containing 5% CO₂. They were routinely tested for mycoplasma infection.

**Cloning.** The *CD19* minigene was amplified from human genomic DNA (Promega) with the primers 5′-catAAGCTTgaccaccgccttcctctctg-3′ and 5′-cat-GAATTCNNNNNNNNNNNNNNNNNNNGGATCCttcccggcatctccccagtc-3′. pcDNA3.1 was used as the vector backbone for the *CD19* minigene plasmid. Both the backbone as well as the minigene amplicons were digested with the restriction enzymes *Eco*RI and *Hind*III (New England Biolabs). The backbone was extracted from a 1% agarose gel using QIAquick Gel Extraction Kit (Qiagen) and the minigene insert was cleaned up using QIAquick PCR Purification Kit (Qiagen). Ligation was conducted overnight at 16 °C with T4 DNA Ligase (New England Biolabs). All minigene mutations were introduced via Q5 Site-Directed Mutagenesis Kit (New England Biolabs). Position 748 (nucleotide 6 of intron 2) was exchanged from G to T to raise the baseline level of exon 2 inclusion in the WT *CD19* minigene to a similar level as in the endogenous *CD19* gene. The nine mutations from eight patients in Orlando et al.[5] listed in Supplementary Table 1 were inserted into the WT *CD19* minigene. For validation experiments (Fig. 3e), 19 individual point mutations with predicted effects on at least one isoform were inserted into a *CD19* minigene variant that additionally contained the mutation G742C to adjust the baseline of splice isoforms in HEK293 cells to the pattern seen in NALM-6 cells. All kits were used according to the manufacturers' recommendations.

**Mutagenesis of minigene and library construction.** For the random mutagenesis of the *CD19* minigene, GeneMorph II Random Mutagenesis Kit (Agilent) was used according to manufacturer's recommendations using 500 ng *CD19* minigene for 30 cycles at 56 °C with the amplification primers 5′-catAAGCTTgaccaccgccttcctctctg-3′ and 5′-catGAATTCNNNNNNNNNNNNNNNNNNNGGATCCttcccggcatctccccagtc-3′. PCR products were purified using QIAquick Gel Extraction Kit (Qiagen), digested with *Eco*RI and *Hind*III (New England Biolabs) and then ligated into the backbone.

**Transfection with minigenes**. Cells were twice washed in Dulbecco's phosphate-buffered saline (DPBS, Gibco Thermo Fisher Scientific) and then collected in R buffer with a density of $2 \times 10^7$ cells/ml. For electroporation, we used 5 µg plasmid DNA (with a concentration of at least 1 µg/µl) to $2 \times 10^6$ cells in R buffer for a 100 µl NEON electroporation pipette tip (Thermo Fisher Scientific) at 1600 V for 30 ms and 1 pulse. Cells were harvested 24 h later. For validation experiments (Fig. 3e), $7 \times 10^5$ cells per well were seeded in six-well TC plates one day prior to transfection. 24 h later, cells were transfected with a mixture of 2 µg plasmid DNA and 20 µg linear Polyethylenimine MW 2500 (PEI 2500, Polysciences), and Gibco Opti-MEM (Thermo Fisher Scientific) was added to 100 µl total volume. This mixture was added dropwise to 1,9 ml fresh DMEM covering the cells, followed by incubation for 24 h.

**Quantification of splicing isoforms using semiquantitative RT-PCR**. Semi-quantitative RT-PCR was used to quantify ratios of *CD19* mRNA isoform variants. To this end, reverse transcription was performed on 500 ng RNA with RevertAid Reverse Transcriptase (Thermo Fisher Scientific) according to the manufacturer's recommendations. Subsequently, 1 µl of the cDNA was used as template for the RT-PCR reaction with OneTaq DNA Polymerase (New England Biolabs). PCRs were run at the following conditions: 94 °C for 30 s, 28 cycles (minigene) or 34 cycles (endogenous *CD19*) of [94 °C for 20 s, 55 °C for 30 s, 68 °C for 30 s] and final extension at 68 °C for 5 min. The primers 5′-ACCTCCTCGCCTCCTCTTCTTC-3′ and 5′-GCAACTAGAAGGCACAGTCG-3′ were used for the *CD19* minigene, and 5′-ACCTCCTCGCCTCCTCTTCTTC-3′ and 5′-CCGAAACATTCCACCGGAA CAGC-3′ for the endogenous *CD19* gene. A TapeStation 2200 capillary gel electrophoresis instrument (Agilent) was used for quantification of the PCR products on D1000 tapes.

For the semiquantitative RT-PCR experiments in HEK293 cells, cells were harvested 24 h after transfection and pelleted. RNA was isolated using the QiaGen RNeasy kit following the manufacturer's protocol with the exception of adding only 100 µl of cell lysate onto the gDNA removal columns to ensure proper removal of genomic and/or plasmid DNA. 1–2 µg RNA per sample were used to generate cDNA with the ThermoScientific RevertAid cDNA kit. We changed the second temperature step of the manufacturer's synthesis protocol from 4 °C to 25 °C (5 min) to further reduce RNA dimerisation or formation of secondary structures. Subsequently, 1 µl of cDNA was used as template for the RT-PCR reaction with OneTaq DNA Polymerase (New England Biolabs). PCRs were run at the following conditions: 94 °C for 30 s, 28 cycles of [94 °C for 20 s, 52 °C for 30 s, 68 °C for 30 s] (minigene) or 34 cycles of [94 °C for 20 s, 55 °C for 30 s, 68 °C for 30 s] (endogenous CD19) and final extension at 68 °C for 5 min. The primers 5′-ACCTCCTCGCCTCCTCTTCTTC-3′ and 5′-GCAACTAGAAGGCACAGTCG-3′ were used for the CD19 minigene, and 5′-ACCTCCTCGCCTCCTCTTCTTC-3′ and 5′-CCGAAACATTCCACCGGAACAGC-3′ for the endogenous CD19 gene. The PCR products were quantified using the TapeStation 4200 system and the High Sensitivity D1000 reagents and tapes (Agilent) according to the manufacturer's protocol.

Significance of differences in isoform abundance for comparing WT minigenes vs. mutated variants (Fig. 1i) or RBP knockdown vs. control (Fig. 5f) was calculated by a Student's *t* test separately for each isoform, reporting the smallest *P* value for each comparison. A one-way ANOVA was used to test whether isoform abundances are different in any of the three conditions shown in Fig. 1g.

## Generation of stable and inducible shRNA knockdown cell lines

*Production and preparation of lentivirus*. Oligonucleotides with shRNA inserts against eleven RBPs (Supplementary Table 2) were ordered as Ultramer DNA Oligos from Integrated DNA Technologies (Leuven, Belgium). All sequences were based on[56]. Oligonucleotides containing shRNA inserts were PCR-amplified with primers 5′-TCTCGAATTCTAGCCCCTTGAAGTCCGAGGCAGTAGGC-3′ and 5′-TGAACTCGAGAAGGTATATTGCTGTTGACAGTGAGCG-3′ and purified with QIAquick PCR Purification Kit (Qiagen). shRNA inserts and miR-E18_LT3GEPIR_Ren714 backbone (inducible via Tet-On system) were cut with *Eco*RI and *Xho*I (New England Biolabs). Backbone was purified from agarose gel with QIAquick Gel Extraction Kit (Qiagen). The fragments were then ligated with T4 DNA Ligase (New England Biolabs) at 16 °C overnight.

Constructs were transduced into NALM-6 via HEK293T-produced lentiviruses. To this end, 10 cm dishes of HEK293T were transfected using 30 µl Lipofectamine 2000 (Thermo Fisher Scientific) with three plasmids: 4 µg shRNA-producing constructs + 2 µg psPAX2 (lentiviral packaging) + 1 µg pMD2.G (lentiviral envelope) at 72 h prior to transduction. On the first day after transfection, the medium was changed. Work with cells used for lentiviral production was conducted in the S2 laboratory.

*Transduction of NALM-6 cells*. Lentiviral production was confirmed with Lenti-X GoStix (Takara) and lentiviruses were concentrated with Lenti-X Concentrator (Takara) according to the manufacturer's recommendations. For transduction, $1 \times 10^6$ NALM-6 cells in 500 µl of medium were added to the concentrated virus. 5 µg/ml polybrene (Sigma-Aldrich) was added. The cells were centrifuged at 800 g and 32 °C for 30 min. Cells were then transferred into 6-well plates and cultivated in normal growth medium without antibiotics. Selection was started after 48 h with 0.5 µg/ml puromycin (Thermo Fisher Scientific). Antibiotic medium was

exchanged every 2–3 days. As soon as cells were not dying under selection anymore and the population was stable, induction experiments were started. After transduction, cells remained in the S2 laboratory for at least 6 weeks. Then, Lenti-X GoStix was used to check for any remaining lentivirus.

*Induction of stable shRNA-expressing NALM-6 cells*. Controlled by the Tet-responsive *TRE3G* promoter, the expression of shRNA was induced by addition of doxycycline (Thermo Fisher Scientific). To this end, $2 \times 10^6$ NALM-6 cells were seeded into a six-well plate in 2 ml medium containing 0.5 µg/ml puromycin and induced with 0.5 µg/ml doxycycline, diluted in RPMI 1640 medium (Thermo Fisher Scientific). Induction was conducted at 37 °C and 5% $CO_2$ and cells were harvested after 48 h. During induction, the shRNA expression system is coupled to the production of eGFP, which was examined by fluorescence microscopy before harvesting.

**Quantitative real-time PCR (qPCR)**. RNA was extracted from the induced harvested cells using the RNeasy Plus Mini Kit (Qiagen). This RNA was used for qPCR to validate the RBP knockdown as well as for semiquantitative RT-PCR experiments to check the splicing pattern of endogenous *CD19*. The qPCR was conducted using the Luminaris HiGreen qPCR Master Mix, low ROX (Thermo Fisher Scientific) according to the manufacturer's recommendations. Oligonucleotide sequences of all qPCR primers are given in Supplementary Table 3.

*Targeted DNA sequencing*. DNA-seq of the minigene library was performed on the PacBio SMRT sequencing platform at MPI-CBG Dresden. For this purpose, the minigene plasmid library was digested with *Eco*RI and *Hind*III (New England Biolabs) and run on an agarose gel. The desired band at the size of 1301 nt was cut out and purified using QIAquick Gel Extraction Kit (Qiagen). For the run on the PacBio SMRT cell, standard library preparation was performed.

*Targeted RNA sequencing*. NALM-6 cells were electroporated with the mutated minigene library (see above). 24 h later cells were harvested and RNA was isolated via the RNeasy Mini Kit (Qiagen). 20 µg isolated RNA was poly-A-selected using Dynabeads Oligo (dT)$_{25}$ beads (Invitrogen) according to the manufacturer's recommendations. Reverse transcription was performed on 500 ng poly-A-selected RNA with RevertAid Reverse Transcriptase (Thermo Fisher Scientific) according to the manufacturer's recommendations. To prevent chimeric amplicons, the RNA-seq libraries were amplified via emulsion PCR[57] using the Phusion DNA Polymerase (New England Biolabs). The following primers containing Illumina adaptors were used in the PCR: 5′- CAAGCAGAAGACGGCATACGAGATCGGTCTC GGCATTCCTGCTGAACCGCTCTTCCGATCTNNNNNNNNNNNGGAACCTCT AGTGGTGAAGG-3′ (fwd) 5′-AATGATACGGCGACCACCGAGATCTACACT CTTTCCCTACACGACGCTCTTCCGATCTNNNNNNNNNNCCGCCAGTGTG ATGGATATC-3′ (rev) under following conditions: 98 °C for 30 s, 25 cycles of [98 °C for 10 s, 63 °C for 20 s, 72 °C for 1 min] and final extension at 72 °C for 5 min. Amplicons were purified using Agencourt AMPure XP beads (Backman Coulter). Purified products were analysed on the TapeStation 2200 capillary gel electrophoresis instrument (Agilent) and quantified using the Qubit assay (Thermo Fisher Scientific). RNA-seq was carried out on the Illumina MiSeq platform using paired-end reads of 350 nt + 250 nt length and a 10% PhiX spike-in to increase sequence complexity.

**PTBP1 iCLIP2 experiments**. We used the iCLIP2 approach for transcriptome-wide mapping of PTBP1 binding to RNA in NALM-6 cells. iCLIP2 was performed according to our previously published protocol[35]. Briefly, the iCLIP2 libraries were made from NALM-6 cells grown in RPMI as described above ($2 \times 10^6$ cells per replicate). To induce protein-RNA crosslinks, the cells were irradiated with 150 mJ/cm$^2$ UV light at 254 nm. Next, PTBP1 protein-RNA complexes were immuno-precipitated using 2 µg of anti-PTBP1 antibody (Santa Cruz, sc-56701). RNase digestion was performed by adding 10 µl of 1/300 or 1/500 diluted RNase I (Ambion) to the sample. RNA purification, reverse transcription and library preparation were done as described in[35].

**PTBP1 siRNA electroporation in MHHCALL4 and P493-6 cells**. The cell lines MHHCALL4 and P493-6 were electroporated with a specific siRNA targeting PTBP1 (TAGCAAGATGATACAATGGTA[dT][dT]; Sigma, sR90) or a Scramble control (D-001810-10-50, Dharmacon) using the Neon Transfection System (Thermo Fisher Scientific). In short, $5 \times 10^5$ cells were resuspended in 10 µl of 5 µM siRNA in buffer R and electroporated using the Neon Transfection System 10 µL Kit (MPK1096, Thermo Fisher Scientific) with the following settings: 1700 V, 20 ms, 1 pulse. After electroporation, the cells were cultured in the recommended media for 48 h and collected for CD19 cell surface staining, quantitative real-time PCR and Western blot.

*CD19 cell surface staining*. In all, $1 \times 10^5$ cells were resuspended in 50 µl of PBS, 20% FBS, 1 mM EDTA and 2.5 µl of Human TruStain FcX blocking (422302, BioLegend) and incubated for 20 min. After blocking, 2.5 µl of APC anti-human CD19 antibody (1:20, 982406, BioLegend) was added to the cells and incubated for 30 min. Cells were washed twice with PBS, 20% FBS, 1 mM EDTA and the CD19 staining was measured using the BD Accuri C6 Plus Flow Cytometer

instrument (BD Biosciences). Flow cytometry data was analysed with FlowJo (version 10.7.2) software.

*Western blot.* Cell pellets were resuspended in RIPA buffer with Protease/Phosphatase Inhibitor (1861282, ThermoScientific) and 30 μg of protein were loaded in a 10% pre-cast gel (456-1035, BioRad). Antibodies against CD19 (1:1000, #3574, Cell Signalling), PTBP1 (1:500, sc-56701, Santa Cruz Biotechnology) and β-Actin (1:5000, 8H10D10, Cell Signalling) were used for total protein expression detection. Images were acquired with GBox instrument (Syngene).

*Quantitative real-time PCR.* RNA was extracted using the Maxwell RSC Instrument (Promega) and the Maxwell® RSC simplyRNA Cells Kit (AS1390, Promega). 0.5 μg or RNA was reverse-transcribed using the SuperScript™ IV Reverse Transcriptase kit (18090010, Invitrogen) following the manufacturer's protocol. RNA expression was measured using SYBR Green Master MIX (Thermo Fisher Scientific). Quantitative real-time PCR was performed in a QuantStudio™ 7 Pro Real-Time PCR System (Thermo Fisher Scientific) with specific primers (Supplementary Table 3) spanning the exon-exon junctions between exons 2 and 3 (E2E3 1&2), 3 and 4 (E3E4), and 10 and 12 (E10E12) as well as the exon-intron junctions from exon 2 to intron 2 (e2i2), from intron 2 to exon 3 (i2e3).

**Re-analysis of RNA-seq data from Orlando et al**. We re-analysed RNA-seq data of B-ALL patients at screening and after CART-19 therapy relapse from Orlando et al.[5] to quantify intron 2 retention in *CD19*. Since raw data were not available, we obtained BAM files for the different patients deposited in the Short Read Archive (SRA) under the accession SRP141691. For 10 patients, matched data were available at screening and relapse. We excluded one patient (patient #17) after visual inspection indicating that the submitted data in fact corresponded to DNA-seq rather than RNA-seq data. The data contained the aligned reads mapped to several genes from the immune system including *CD19*. Using custom scripts, we extracted the sequence of the reads, reformatted them and generated fastq files. We then mapped the fastq files to our minigene sequence using STAR[58] (v2.6.1). We used the re-mapped reads to quantify the levels of intron 2 retention in the different samples using the R/Bioconductor package ASpli[59] (version 1.12.1).

For the expression analysis of B-ALL patients at screening and after CART-19 therapy relapse, we used the gene read counts provided in Supplementary Data Table 1 of Orlando et al.[5]. Gene lengths were taken from BiomaRt (version 2.4.21) for the human genome version GRCh37 accessed through the R/Bioconductor package OrgDb (version 3.10.0). Normalisation and RPKM calculations were performed using the R/Bioconductor package edgeR[60] (version 3.28.1).

**DNA-seq barcode demultiplexing**. We obtained the circular consensus sequences (CCS), stored as fastq files. Two rounds of sequencing yielded a total of 337,215 CCS. We kept only reads with a length of 150–1150 nt. We adapted the demultiplexing procedure described in[19]. In this case, we searched for the 15-nt barcode in the last 50 nt of the read. If the barcode was not found, we searched in the last 50 nt of the reverse complementary strand. We only allowed the recovery of barcodes ranging from 14 to 16 nt, which would account for barcodes containing one nucleotide inserted or deleted. Before proceeding with the variant calling, we determined a cutoff to decide the minimal number of CCS to call variants on. Here, we kept only barcodes supported by at least 4 CCS. In total, we recovered 68.5% of all the demultiplexed barcodes which corresponded to 10,558 different minigenes, closely resembling the ~10,000 minigene clones that were used to generate the library.

**DNA-seq mapping and variant calling**. We use BLASR[61] with the standard parameters to map the demultiplexed minigene sequences to the minigene reference. We performed variant calling in the aligned BAM files using the GATK[62] Haplotype-peCaller (version 4.0.10) with the parameters --kmer-size 10 --kmer-size 15 --kmer-size 25 --allow-non-unique-kmers-in-ref. We used different *k*-mer sizes to improve the detection of problematic regions. Mixed barcodes, i.e., barcodes containing two classes of mutations, were removed based on the "penetrance score", reported as allele frequency (AF) in the GATK vcf output files, such that barcodes with more than 25% variants of low penetrance (AF < 0.8) were discarded. Using this strategy, we were able to recover 100,135 mutations of high quality coming from 10,295 distinct minigenes plus an additional 194 unmutated WT minigenes with distinct barcodes. 57.4% of the mutations appeared in at least ten different minigenes.

**RNA-seq barcode demultiplexing**. RNA-seq libraries were sequenced on Illumina MiSeq as 350 nt + 250 nt paired-end reads, yielding approximately 23 million reads. We controlled their quality using FastQC (version 0.11.5, https://www.bioinformatics.babraham.ac.uk/projects/fastqc/) and removed bad quality ends of reads using Trimmomatic[63] (version 0.36, parameters: SLIDINGWINDOW:6:10 MINLEN:0). After trimming, we filtered for read pairs with a minimal length of 305 nt (read1) and 157 nt (read2) and, as done in Braun et al.[19], we used matchLRPatterns() and trimLRPatterns() from the R/Bioconductor package Biostrings to extract the 15-nt barcode in read1 between the two flanking restriction sites (Lpattern = TGCAGAATTC, Rpattern = GGATCC) allowing one mismatch. All read pairs with barcode length between 14 and 16 nt were kept for further processing. Barcode sequences were added to the read names in the fastq file and 5′

ends of reads were trimming (read1: everything until the second anchor sequence GGATCC, read2: the first 12 nt). After identifying and trimming the barcode and other regions, we used Cutadapt[64] (version 1.6, parameters: --adaptor=TA-GAGGTTCC --overlap=3 --error-rate=0.1 --no-indels --minimum-length=244 --pair-filter=both) to remove remaining primer sequences from read1. Lastly, the barcode information attached to the read names was used to demultiplex all read pairs into individual fastq files for each minigene.

**Isoform quantification from RNA-seq data**. Only barcodes/minigenes also detected in the DNA-seq library were kept for further analysis. All minigenes with insertions or deletions of 10 or more base pairs were removed from further analysis. For better mapping results, we shortened read1 to at most 260 nt. Read pairs of each minigene were mapped to the respective minigene (including all mutations with penetrance ≥0.8, but excluding insertions and deletions) using STAR[58] (version 2.6.1b). An annotation of three isoforms (exon 2 inclusion and skipping, as well as the artefact PCR product Δex2part which lacks an internal fragment of exon 2 due to a reverse transcription artefact[65]) was provided to STAR during mapping and an --sjdbOverhang of 259 was set. When running STAR, all SAM attributes were written, up to ten mismatches were allowed, soft-clipping was prohibited on both ends of the reads and only uniquely mapping reads were kept for further analysis. BAM files were sorted and indexed using SAMtools[66] (version 1.5).

Properly and consistently mapped pairs were used for isoform reconstruction using a custom Perl script. Read pairs were considered properly mapped if they mapped with the right orientation on opposite strands. Read pairs mapped consistently if they either did not overlap or in case of an overlap, agreed in their detected splice junctions. Besides, only read pairs for which both mates exceeded the constitutive exon boundaries by at least 10 nt were used for isoform reconstruction. All other pairs were removed since they did not provide any isoform information. Only minigenes covered by at least 100 read pairs usable for isoform reconstruction were kept for further analysis. For each read pair, the CIGAR strings of the two mates were used to reconstruct their splicing isoform. Regarding the artefact product Δex2part, we combined the eight possible mappings of the missing internal fragment of exon 2 which are possible due to the associated 8-nt repeat sequence[65]. Only isoforms, which are supported by ≥ 1% of the read pairs and at least two read pairs in at least one minigene, were kept for further analysis.

The analysis described above was done separately for two replicates. All isoforms occurring with a frequency of at least 5% in two or more minigene variants in either of the two replicates were kept as individual isoforms. All other detected isoforms were summarised into a category "discarded". Isoforms with Δex2part, i.e., excluding the internal intron in exon 2, were combined with their "real" counterparts without Δex2part by merging isoforms that only differed in the exclusion of the internal fragment of exon 2. In total, this leads to a set of 101 individual isoforms.

**Estimation of single-mutation effects and splicing-affecting mutations**. Since the majority of the minigenes in the dataset exhibit more than one mutation, with a mean of 9.6 mutations per minigene, the splicing-affecting mutations cannot be read out directly from the data. We used multinomial logistic regression to infer the effects of single mutations from combined measurements. The regression is based on hypothetical minigenes containing only one mutation, and on the assumption that mutation effects (log fold-changes compared to WT) add up into combined ones at the levels splice isoform ratios[19].

For regression, we focused on the five major isoforms that are already present in the WT minigene (see main text). Therefore, minigenes exhibiting more than 5% cryptic isoforms were removed from the dataset, and for the remaining minigenes the cryptic isoforms were merged into a lumped splicing category which we termed "other". Thus, six categorical splicing outputs (inclusion, skipping, intron2-retention, alt-exon2, alt-exon3, other) were considered in the regression model, and the probability of each these outputs to be observed was assumed to equal the measured isoform frequencies. The regression was formulated as a softmax regression problem using the LogisticRegression command from the Python package scikit-learn[67].

Given the large number of mutations per minigene in the dataset, the regression was prone to overfitting (i.e., mutations with weak effects on splicing were assigned non-zero coefficients to fit random fluctuations in the data; not shown). To avoid this problem, we employed L1 penalisation. The strength of the penalty was optimised by tenfold cross-validation, and the resulting inverse regularisation strength was C = 10 for both replicates.

The goodness of the model in describing the measured combined mutation effects (minigenes) was tested by assessing the correlation between model and data in training and test datasets (Supplementary Fig. 4a). Tenfold cross-validation was performed by once randomly splitting the dataset into ten parts. In ten distinct model evaluations, nine of the data sections were simultaneously used for model training, whereas the remaining data section served as test data. Therefore, each data point is only once part of the test data, and the mean Pearson correlation coefficient between model prediction and test data was used to assess the model performance. Cross-validation at the final penalisation strength (with the highest correlation between model and test data) showed that the method performs very well in estimating the minigene isoform frequencies of the test dataset

(Supplementary Fig. 4b). Since we saw little variability in the prediction power for these ten validation runs, we report the average correlation coefficient in Fig. 3b: The Pearson correlation coefficients between softmax predictions of combined mutation effects and measurements lie for the single isoforms between 0.68-0.95 for the first replicate and between 0.71-0.93 for the second replicate.

The accuracy of the model-predicted single-mutation effects in the softmax regression was assessed by leaving out 56 directly measured single-mutation minigenes (i.e., minigenes bearing only one mutation) from the training data. Since most of these 56 mutations are not splicing-affecting, we focused our analysis on the seven mutations that change the inclusion isoform level beyond two standard deviations of the WT minigene distribution: For each of the seven mutations, we performed multiple softmax fits in which the training data: (i) contained all minigenes not harbouring the mutation of interest, (ii) excluded its single-mutation minigenes, and (iii) comprised varying numbers of combined mutation minigenes containing the mutation. For each mutation occurrence between 1 and 10, we used up to 7 different, randomly chosen combinations of multiple mutated minigenes including the mutation of interest. For each of these models, we generated predictions for the single-mutation effect. The prediction accuracy was assessed by calculating the difference between model and direct single-mutation measurements for a certain mutation occurrence. The standard deviation of the difference between model and data was used as a measure for the model error. We find that a mutation occurrence of 3 leads to an error level equal to two WT standard deviations (calculated based on inclusion levels of all WT minigenes in the first replicate). For higher mutation occurrences, the prediction accuracy does not improve further (Supplementary Fig. 4c).

The final modelling step was to identify splicing-affecting mutations. For this purpose, we adopted an approach analogous to empirical $P$ values, i.e., we compared predicted single-mutation effects to empirical isoform frequency distributions in the WT. Isoform frequencies were measured for 195 and 194 WT minigenes in the two replicates. For each isoform and replicate, we chose the 2.5% and 97.5% quantiles of the respective empirical WT frequency distribution as cutoffs (corresponding to a two-sided 5% cutoff). A mutation was considered to have an effect on a splice isoform if, for both replicates, the frequencies predicted by the model were beyond the respective cutoffs and if the effects were in the same direction.

**Splice site characterisation**. Splice site usage for a given position represents the frequency of the isoforms using a given splice site in a particular minigene divided by the sum of all isoform frequencies for the same minigene. For Fig. 4a, we used the maximum usage of a particular splice site across all minigenes. The strength of putative splice sites along the minigene was calculated using MaxEnt scores[26] in sliding windows of 9 nt or 23-nt to evaluate the corresponding sequences as potential 5′ or 3′ splice sites, respectively. The procedure was repeated for all individual point mutations to assess their potential to create cryptic splice sites. For the calculations, we used the Python implementation of MaxEnt (maxentpy, version 0.0.1, https://github.com/kepbod/maxentpy). We filtered the output by keeping only windows that contained a GU or AG dinucleotide in the positions 4–5 (5′ splice site) or 19-20 (3′ splice site), respectively.

We compared the effects of single-point mutations in our library to predictions by the state-of-the-art deep learning algorithm SpliceAI[25]. We ran SpliceAI (version 1.3.1) with the default parameters plus masking (-M1), using GENCODE[68] (v31) annotation for the human genome version hg38 as a reference. Given that SpliceAI results are reported in terms of a probability of gain or loss of a particular splice site, we assigned the gained splice sites in our cryptic isoforms by comparison to the canonical exon 2 inclusion isoform, such that if a new splice site appears in the cryptic isoform, it is considered as "gained" with respect to the "lost" WT splice site. All splice sites in a cryptic isoform were given the same prevalence score, i.e., the prevalence score of the mutation-isoform pair. To compare the SpliceAI scores for a given splice site gain with our prevalence score (Fig. 4f), we considered the mutations that (i) share the same gain-loss pair of positions in both assays, and (ii) are predicted by SpliceAI to gain of a new splice site (i.e., a cryptic site where score_gain > score_loss) upon a given mutation.

**RBP binding site predictions**. For the prediction of RBP binding motifs, we used the web versions of the oRNAment[28] (http://rnabiology.ircm.qc.ca/oRNAment) and ATtRACT[27] (https://attract.cnic.es/) databases to query the minigene sequence for presence of RBP motifs (Supplementary Fig. 7a). From the obtained predictions, we collapsed overlapping binding sites from the same tool and RBP.

We used DeepRiPe[30] to predict the potential impact of single-point mutations on RBP binding. To this end, we downloaded the trained models for PAR-CLIP and ENCODE eCLIP data on 159 RBPs available in the GitHub repository (https://github.com/ohlerlab/DeepRiPe). We scored each mutation (annotated with regards to the hg38 reference genome) across the individual RBP models and reported every mutation for which the model score changed by at least 0.1 in either direction compared to the WT sequence (Supplementary Data 6, worksheet "DeepRiPe mutations"). Positive and negative delta scores refer to a predicted increase or reduction in RBP binding, respectively. The scoring functions are based on the iPython notebooks provided by DeepRiPe: https://colab.research.google.com/drive/18yeqRE7KmOjfbUaLAfJ6rMBjAulYo-Uc?usp=sharing

For the definition of significant RBP binding sites, we used the following strategy. For binding sites predicted by oRNAment and ATtRACT, we first checked their

overlap separately for each isoform. If a binding site overlapped in at least one position with a splicing-affecting mutation with respect to this particular isoform, we defined this binding site as an isoform-specific significant binding site. All binding sites that were significant for at least one isoform were collapsed into the complete list of significant binding sites, yielding a total of 315 significant binding sites for 74 RBPs. In the case of DeepRiPe, a mutation with a delta score >0.25 for a given RBP model was required to overlap with a splicing-affecting mutation for a particular isoform (our screen) to be considered an isoform-specific significant RBP-changing mutation. In a similar manner, all isoform-specific mutations for any isoform were collapsed into a complete list of significant RBP-changing mutations, yielding a total of 222 significant mutations that affected the binding of 58 RBPs.

**iCLIP data processing**. iCLIP libraries were sequenced on an Illumina NextSeq 500 sequencing machine as 92 nt single-end reads including a 6 nt sample barcode as well as $5 + 4$ nt unique molecular identifiers (UMIs). Basic quality controls were done with FastQC (version 0.11.8) (https://www.bioinformatics.babraham.ac.uk/projects/fastqc/) and reads were filtered based on sequencing qualities (Phred score) in the barcode region using the FASTX-Toolkit (version 0.0.14) (http://hannonlab.cshl.edu/fastx toolkit/) and seqtk (version 1.3) (https://github.com/lh3/seqtk/). Reads were demultiplexed based on the experimental barcode, which is found on positions 6 to 11 of the reads, using Flexbar[69] (version 3.4.0). Afterwards, barcode regions and adaptor sequences were trimmed from read ends using Flexbar. Here, a minimal overlap of 1 nt of read and adapter was required, UMIs were added to the read names and reads shorter than 15-nt were removed from further analysis. Downstream analysis was done as described in Chapters 3.4 and 4.1 of Busch et al.[70]. Genome assembly and annotation of GENCODE[68] v31 were used during mapping.

**Patient data analysis**. RNA-seq data of 222 B-ALL patients from the Therapeutically Applicable Research To Generate Effective Treatments (TARGET) programme (https://ocg.cancer.gov/programs/target) were processed from fastq files. Sequencing adaptors were trimmed with TrimGalore[71] (version 0.6.6), aligned to the hg38 human genome assembly with STAR[58] (version 2.5.2a), and sorted and indexed with SAMtools[66] (version 1.11). Splice junctions were quantified individually for each sample using MAJIQ[72] (version 2.2) and ENSEMBL reference transcriptome GRCh38.94[73]. Only splice junctions with a usage level (percent selected index, PSI) of at least 5% in any given TARGET B-ALL samples were quantified. The local splicing variation (LSV) harbouring alternative splicing in the region of the *CD19* minigene (Supplementary Fig. 1a) was quantified in 220 out of 222 B-ALL patients. For comparison, we used RNA-seq data of immature and mature B cells from healthy donors from[44,74].

Annotated variant call format (VCF) files were downloaded for the TARGET B-ALL patient cohort from the NCI Genomic Data Commons (GDC) Data Portal (accessed 11/18/2021). These files are available under controlled access (see Acknowledgements). In brief, the VCF files had been generated from patient whole-exome DNA-seq data using the GDC DNA-seq Analysis Pipeline (https://docs.gdc.cancer.gov/Data/Bioinformatics_Pipelines/DNA_Seq_Variant_Calling_Pipeline/) which includes genomic alignment with BWA, data clean-up with Picard tools and GATK, and calling of somatic variants from matched samples of tumor and adjacent normal tissue for each patient with Mutect2. The raw VCF files were further annotated using the Variant Effect Predictor (VEP) tool to infer the location of each mutation, its consequence (frameshift/silent mutation) and the affected gene(s) as well its overlap with known variants in databases such as ClinVar and dbSNP. Using custom scripts, a total of 468 VCF files of TARGET B-ALL patients were parsed for *CD19* mutations. This identified 39 patients with somatic mutations within the *CD19* gene, including 11 mutations in the *CD19* minigene region (Supplementary Data 4).

**Reporting summary**. Further information on research design is available in the Nature Research Reporting Summary linked to this article.

## Data availability
The sequencing data generated in this study have been deposited in the Gene Expression Omnibus (GEO) database under accession code GSE182894. The collection consists of the PacBio DNA-seq libraries (GSE182892) [https://www.ncbi.nlm.nih.gov/geo/query/acc.cgi?acc=GSE182891], the Illumina RNA-seq libraries (GSE182892) and the PTBP1 iCLIP2 libraries in NALM-6 cells (GSE182893). The results published here are in whole or part based upon data generated by the Therapeutically Applicable Research to Generate Effective Treatments (https://ocg.cancer.gov/programs/target) initiative, phs000218. The data used for this analysis are available at https://portal.gdc.cancer.gov/projects. The remaining data are available within the Article, Supplementary Information or Source Data files. Source data are provided in this paper.

## Code availability
The computational code for the analyses and figure generation is available in Zenodo [https://doi.org/10.5281/zenodo.6614454]/Github [https://github.com/mcortes-lopez/CD19_splicing_mutagenesis] under an open-source MIT license.

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

## Acknowledgements

The authors would like to thank the members of the participating labs for their support and discussion. We would like to thank Sylvia Weiss, Dpt. Systems Biology, University of Stuttgart for technical assistance. We gratefully acknowledge the Institute of Molecular Biology Core Facilities for their support, especially the Genomics Core Facility and the use of its NextSeq 500 (funded by the Deutsche Forschungsgemeinschaft [DFG, German Research Foundation] INST 247/870-1 FUGG) and the Bioinformatics Core Facility. We gratefully acknowledge the PacBio SMRT sequencing platform at MPI-CBG Dresden. TARGET data used for the analyses were accessed under dbGaP project #10088: "Alternative splicing in pediatric cancers", sub-studies phs000463.v21.p8 and phs000464.v21.p8 (Acute Lymphoblastic Leukemia (ALL) Pilot Phase 1 and Expansion Phase 2). This work was funded by the Naturwissenschaftlich-Medizinische Forschungszentrum (NMFZ) to J.F., J. K. and C. P. and the Deutsche Forschungsgemeinschaft (DFG) to K.Z., J.K. and S. L. (ZA 881/2–3 to K. Z., KO 4566/4-3 to J. K., and LE 3473/2–3 to S. L.). K. Z. was also supported by the Deutsche Forschungsgemeinschaft (SFB902 B13). This work was supported by the grant from the National Institutes of Health (U01 CA232563 to A. T.-T. and Y. B.), St. Baldrick's-Stand Up to Cancer (SU2C-AACR-DT-27-17 to A. T.-T.) and the V Foundation for Cancer Research (T2018-014 to A. T.-T.). M. T. D. acknowledges support from The Ellen Weisberg Fund: Advancing Breakthroughs in Pediatric Cancer.

## Author contributions

M.C.-L. performed most bioinformatics analyses. L. S. performed the *CD19* minigene experiments as well as the massively parallel *CD19* splicing reporter assay. L.S. and B.S performed shRNA-mediated RBP knockdown experiments and corresponding splicing assays. M.M. and M.M.M. helped with experiments. M.E. and S.L. designed the mathematical modelling and prevalence score approach, and M.E. performed the analyses. F.K. contributed to the quantification of mutation effects. S.U. and M.K. validated single-mutation effects from model predictions. A.O., M.C.-L., L.S. and J.K. performed PTB iCLIP experiments. M.T.D. performed CD19 flow cytometry assays upon *PTBP1* knockdown and associated measurements. A.B. performed iCLIP and RNA-seq data processing as well as splice isoform quantification. M.Q.-V., M.T.D. and R.S. performed TARGET ALL data analysis under supervision of Y.B. and A.T.-T.. Study was designed by M.C.-L., L.S., M.E., K.Z., S.L. and J.K. with help from C.P., J.F. and all co-authors. K.Z., S.L. and J.K. supervised most of the bioinformatics analyses, mathematical modelling, and experimental work, respectively. M.C.-L., L.S., M.E., C.P., K.Z., S.L., and J.K. wrote the manuscript with help and comments from all co-authors.

## Funding

## Competing interests

A.T.-T. has an interest in intellectual property "Discovery of CD19 Spliced Isoforms Resistant to CART-19". This interest does not meet the definition of a reviewable interest under Children's Hospital of Philadelphia's (CHOP's) conflict of interest policy and is therefore not a financial conflict of interest. Furthermore, this intellectual property has not been licensed or otherwise commercialised to date. However, should this technology be commercialised in the future, A.T.-T. would be entitled to a share of royalties earned by CHOP per its patent policy. The other authors have no competing interests.
