## [Peer Review File · Nature Communications]

High-throughput mutagenesis identifies mutations and RNA-binding proteins controlling CD19 splicing and CART-19 therapy resistanceREVIEWER COMMENTS

Reviewer #1 (Remarks to the Author): with expertise in CD19 CAR-T

In this manuscript, Cortés-López et al. exploit a splicing reporter assay and a CD19 minigene to uncover (in an unbiased fashion) mutations in CD19 gene (from exon1 to exon3) that give rise to various CD19 isoforms. Using a mathematical approach, they assigned DNA mutations to protein isoforms. This study also suggests that some mutations can alter binding sites for RNA binding proteins (RBP) involved in mRNA splicing. Following an in silico screen, 11 RBP candidates were shortlisted and knocked-down in NALM-6 cell line. Finally, the authors uncovered that depletion of some RBP proteins and notably PTBP1 yields to non-functional CD19 isoform.

This work is well performed and contributes to the understanding of mechanisms leading to the alteration of CD19 expression, which is of significance to the field of CART-19 therapy. It also represents a helpful resource for CD19 mutations/isoforms.

I believe that this study will have more impact if the authors address the following points:

1) This manuscript is dedicated to CD19 mutations and their impact on CART-19 therapy, thus I expect the authors to go beyond an exhaustive list of mutations and provide a clearer answer as to how CD19 protein is lost in B-ALL:

- Figure 1 A & B. The authors observed an increase of Intron2-retention upon CART-19 therapy relapse. Before CART-19 therapy (so cells are supposed to be CD19+), Intron2-retention is clearly detected. This cannot result from the RNA contribution of rare CD19neg clones that pre-exist before CAR-T treatment, likely the aberrant isoform is already present in CD19+ B-ALL cells. What is the scenario proposed by the authors? Is there a monoallelic alteration in CD19+ B-ALL? Then a loss of heterozygosity upon treatment? Alternatively, do they envision a more efficient mis-splicing in CD19neg B-ALL (due to downregulation of a trans-acting factor such as PTBP1)?

- Fig 1B. One B-ALL sample displays ~100% Intron2-retention frequency at both 'initial screening' and upon CART-19 relapse. What was the expression level of CD19 protein in this B-ALL sample before treatment? (I guess the B-ALL was initially CD19+, how this can be consistent with a 100% Intron2-retention frequency?)

-Strikingly, the manuscript starts (fig 1 A &B) and finishes (fig 5H & I) with the same 'idea': Intron2-retention is largely present in B-ALL samples. Fig 5H & I show that 93% of B-ALL possess more than 50% of Intron2-retention junction. With such numbers, why aren't CD19neg B-ALL more prevalent (even without anti-CD19 therapeutic pressure)?

2) Is Intron2-retention detected in non-tumoral B cells?

3) Are the mutations found at the relapse (figure S1) detected before CAR-T therapy?

4) The authors indicate that the knockdown of PTBP1 (and in lesser extent PCBP2 and SF3B4) in NALM-6 cell line favours Intron2-retention and they showed in figure S5, that genes corresponding to these proteins are expressed in B-ALL. They should also show the expression of these genes in relapsed B-ALL. Especially, they should compare the expression level of PTBP1 in relapsed B-ALL samples harbouring Intron2-retention versus samples before treatment.

In this context, the authors could also complement figure 5H and show the expression of PTBP1 (and other RBP genes of interest) in the B-ALL TARGET cohort.

5) Concerning mathematical modelling, it seems that the model is less robust for the prediction of intron2-retention and alt-exon2 (fig 3B and S3B). Can the authors comment on that?

6) The authors uncovered that CD19 exon 1-3 minigene can generate several cryptic isoforms. Yet, they did not show whether some of those described isoforms really exist in B cells/B-ALL primary samples.

7) The authors should describe the PTBP1 iCLIP2 assay.

8) DeepRiPe predicts changes of RBP binding upon DNA mutations. If I well understood only altered ('knocked-out') mutations are considered. Have the authors tried to do the analysis the other way round: by defining which mutations result in a knock-in of a RPB binding site?

Reviewer #2 (Remarks to the Author): with expertise in alternative splicing, cancer

In this paper, Cortés-López et. al. study how aberrant alternative splicing of CD19 leads to loss of cognate CD19 epitope necessary for success of CART-19 immunotherapy for management of B-cell acute lymphoblastic leukemia. Focusing specifically on CD19 exons 1-3, the authors utilize a minigene system to conduct a high-throughput mutagenesis screen. In conjunction with quantitative modelling using multinomial logistic regression, the authors identified novel intronic and exonic cis-regulatory mutations that give rise to alternative isoforms that encode for non-functional CD19 proteins. Furthermore, through in silico analyses and knockdown of RNA-binding proteins (RBPs), the authors characterized trans-regulators that promote production of CART-19 sensitive isoforms. While there are sections where the connection between experimental results to patient data is strenuous, the authors, utilize rigorous bioinformatic approaches to offer insight on a novel mechanism of treatment resistance. Further experiments are needed to strengthen their conclusions.

Major comments:

1. Fig. 1. Patient mutations cause splicing changes in the CD19 minigene. Except for mutations from patients 4, 5, 14, and 15, it is not clear that the isoform frequencies caused by the mutations observed in other treatment refractory patients are significantly different from the wildtype control. Inclusion of statistical tests will be helpful here. Furthermore, do the splicing patterns observed in the minigene (Fig. 1F) match that of the patient data (Fig. 1B)?

2. Fig. 2. The accuracy of the screen can be strengthened by a subset of RT-PCR validations.

3. Fig. 3. In the cross-validation, is the 10% that is left out never seen by any of the validation process? Or is a different 10% left out in each validation? This clarification is important as in the latter case, the validation result is influenced by the modeling from the same cohort and is not considered as a true validation.

4. Fig. 4. Mutations generated in the high-throughput mutagenesis screen gives rise to cryptic isoforms. Are any of the mutations from the high-throughput assay overlap with known mutations in B-ALL patients?

5. Fig. 4C. CRISPR replacement of the mutant sites should be performed to indicate that mutations cause the change of splicing and loss of CD19 protein production on cell surface. This will be needed for the conclusion that high-throughput mutagenesis identifies mutations controlling CART-19 therapy resistance.

6. Fig. 5. Do PTBP1 KD cells show decreased CD19 protein on cell surface? This information is needed to connect PTBP1's regulation on CD19 splicing and CART-19 therapeutic resistance. Moreover, it is not clear to the reviewer whether CD19 intron retention causes CART-19 therapy resistance. Are the functional domains deleted when the intron is retained? Were there reports to show causality of CD19 intron retention in therapy resistance?

7. Fig. 5 H. The authors analyzed RNA-seq data of 220 B-ALL patients. Are there CD19 mutations matching the known mutations or the high-throughput results? What are the levels of PTBP1? Do they show correlations with the levels of CD19 intron retention? A significant correlation would serve as a separate validation supporting the premise of the study. An absence of correlation could suggest that other mechanisms rather than mutations and PTBP1 cause CD19 intron retention.

8. The observation in Fig. 5H that all B-ALL patients show CD19 retained intron from 30% to nearly 100% suggest that CD19 intron retention is commonly observed in B-ALL. Does CD19 show retained intron in normal B cells? Results in Fig. 5H imply that the level of intron retention determines therapy resistance and not its presence since all patients showed intron retention. Is this supported by clinical data?

Minor comments:

1. Fig. 1. The authors state that the minigene generates the same isoforms as the endogenous CD19 gene. However, in both gel-like representation and quantification of semi-quantitative RT-PCR, there appears to be higher quantity of the intron2-retention isoform in the minigene construct. Inclusion of p-value for the quantification would be beneficial.

2. Fig. 1D. The difference in splicing pattern observed in K562 cells can be cell type specific. It cannot be compared with NALM-6. The reviewer recommends elimination of K562 results.

Reviewer #3 (Remarks to the Author): with expertise in alternative splicing, RPB

The publication by Cortes-Lopez, Schulz, Enculescu et al. describe a deep dive into the regulatory landscape of CD19 exon 2 processing, including detailed mutational analysis and mathematical modeling to characterize the landscape of complex splicing mis-regulation that can occur with this event. I think this is both quite interesting not only biologically (as mis-splicing of CD19 exon 2 is important physiologically, as well-described in the text), but also as a great model for how such detailed analysis of an individual event can reveal unexpected complexity in splicing regulation (and how different experimental and modeling approaches can be used to understand the mechanisms behind this complexity).

The only significant issues I would raise are:

1) Since they are based off of actual publicly available CLIP datasets, I find it somewhat peculiar to only use the abstracted 'predicted binding sites' for analysis in Figure 5; although I think the analyses presented are well-described, it seems to me to be fairly obvious (and easy) to also calculate and show these overlaps using the actual ENCODE & PAR-CLIP datasets (particularly for the RBPs shown in Fig. 5D, and particularly as many of them (TAF15, SF3B4, PUM2, PTBP1, HNRNPM, HNRNPK, and FUS, based on a quick scan of the ENCODE website) have data in K562, which seems like it could be reasonably similar to the B-ALL sample type under study here).

2) I think the conclusions PTBP1 section of Fig. 5 should either be written less strongly, or requires additional data and analyses (I'd recommend the former). In particular, to me the DeepRiPe sites in Fig. 5G shows little overlap with the strongest region of iCLIP2 crosslink events, and the ATTRACT sites while overlapping that region also show a large number throughout the entire event (raising questions as to whether they have such a high false positive rate with the PTBP1 motif to be uninformative), so I don't know that I agree that the data presented supports the conclusion "The broad binding at splicing-effective positions and beyond supports that PTBP1 is a <<direct and central

regulator>> of CD19 alternative splicing, with most prominent effects on intron 2 retention.” (emphasis added by me)

3) It would also be helpful throughout to add some additional background rates for comparison throughout. For example, line 284-286 describe the fraction of mutations in close proximity to (real or cryptic) splice sites – what is the background rate for all nucleotides in this region and how enriched is this?

4) I would recommend being more explicit with some conclusions – e.g. ‘Taken together, our results strongly suggest that CD19 mutations contribute to CART-19 therapy resistance by inducing splicing changes’ – unless I’m mistaken, the contribution to CART-19 therapy resistance is entirely inferred in this paper (based on predictions of whether the splicing change observed would create either a loss of exon 2 or frameshift, and based on the assumption that the impact of the mutation observed in the minigene reporter will be recapitulated in the full transcript). It would be more correct to say something like ‘our results indicate that far more CD19 mutations are predicted to create isoforms that would escape CART-19 recognition’

Minor comments:

- I’m confused by the term ‘(near-)constitutive exons like CD19 exon 2’ (line 422) – from the author’s data (and a quick skim of K562 and GM12878 data on the UCSC browser), I wouldn’t refer to this as near-constitutive (as intron 2 retention in particular seems relatively common)

Reviewer #4 (Remarks to the Author): with expertise in alternative splicing, bioinformatics

“High-throughput mutagenesis identifies mutations and RNA-binding proteins controlling CD19 splicing and CART-19 therapy resistance” by Mariela Cortés-López et al., provides a detailed investigation of the CD19 (and exons 1-3 in particular) using high-throughput mutagenesis, mathematical modelling and RBP knockdowns to understand the nucleotides, cis-elements and binding sites that regulate splicing, and the complex isoforms generated by mutations in these elements.

The manuscript provides two contributions. Firstly, the study provides a detailed dissection of the impact of mutations on splicing which is often complex and difficult to predict. By performing a high-throughput mutagenesis screen the authors are able to dissect the contribution of nucleotides to splicing and their impact on the resulting gene architecture. This analysis showed that splice mutations can markedly impact isoform diversity, particularly at exons with weakly competitive splice sites. Accordingly, this provides a systems-level understanding of the splicing code and the impact of mutations on splicing diversity (which is remarkably large).

The second strength of the study is the clinical relevance of the CD19 gene analysed. Mutations to CD19 drive resistance in BALL patients to CAR-T cell therapy. Understanding the role of splicing mutations to this gene may ultimately identify mechanisms to prevent resistance and, more immediately, the data may provide a resource for the interpretation of mutations in BALL patients and provide prognostic markers of CART-19 therapy resistance.

More broadly, the study is well designed, and the manuscript is well-written, with clear figures, rigorous analysis, and fair interpretation of results. The methods are detailed, with data and script appropriately available. I congratulate the authors on the study.

Major points.

1. I do not have any major concerns with the study designs, analysis or interpretation. However, one

suggestion on how the study may be improved would be to provide a greater context for the splicing and expression of the CD19 gene in healthy and BALL patient populations. This would largely involve an analysis of CD19 gene splicing in publicly available RNA-seq data from healthy RNAseq datasets (such as GTex) and from B-ALL patients from the Therapeutically Applicable Research To Generate Effective Treatments (TARGET) program. Given the authors have identified a diversity of splicing junctions using long-read data, these would form useful annotations against which to analyse publicly available short-read data for alternative isoforms that have been otherwise missed.

I realise that this has been performed to varying degrees in some previous studies, and this may be why the authors have not specifically focused on this analysis (indeed the authors present some of this data in the manuscript (such as Figure S6D), however, I believe that foregrounding this analysis would provide the readers with an understanding of the CD19 landscape, and provides a useful context in which to consider the suitability of the CD19 mini-gene assay. This includes how well the CD19 recapitulates healthy and patient splicing (quantitatively and splicing structure)?, as well as interpret the outcomes, including the impact of mutations and resulting alternative isoforms. Attempting to recapitulate the impact of these mutations and their complex splicing outcomes in vivo (using gene editing etc.) would be ideal, however, I realise that this is a large undertaking and outside the scope of this current study.

2. One notable difference is the BALL patients from Orlando study often harbor more complex mutations (deletions or insertions greater than 5nt in length, Table S2), whilst my understanding is that the error-prone PCR generates smaller single-point mutations. It would be helpful to provide a comparison of the mutations (type and quantity) for (i) Orlando study, (ii) Orlando study (iii) within healthy populations (gnomAD) and (iv) within BALL and cancer patient populations (COSMIC, ClinVar etc.).

3. The authors show the SpliceAI predictions correlate relatively well with the outcomes from the high-throughput mutagenesis study and suggest their data can be used to benchmark tools for splicing detection. However, this correlation also suggests that there may be value in performing a more detailed analysis of SpliceAI predictions across the broader CD19 gene (beyond exon 2). Whilst these predictions aren't as rigorous as the mutagenesis assay, they could nevertheless provide a broader landscape in which to interpret mutations that impact CD19 splicing that may drive CAR-T cell resistance. For example, a 'predicted' set of spliceAI elements across the CD19 gene could be similarly analysed with respect to publicly available RNA-seq data and mutation databases (from healthy and BALL patients).

Minor points. The manuscript is very well written, and the figures are clear. I have only a few suggested minor grammatical revisions;

1. Several studies reported that in 40-60% of cases the cancerous B-cells become invisible to the CARTs due to loss of detectable CD19 epitope (CD19-negative)
2. Taken together, our dataset is a comprehensive resource for prognostic markers of CART-19 therapy resistance and for a systems-level understanding of the splicing code.
3. Altogether, the in silico predictions suggest the presence of an extensive RBP network controlling CD19 splicing that may impact on the CART-19 therapy success.
4. Moreover, an upregulation of PTBP1 has been implicated in the acquired resistance of pancreatic ductal carcinoma cells to the chemotherapeutic drug gemcitabine
5. Thus, besides the regulation of protein expression, other factors like cellular availability may further impact on PTBP1 function in B-ALL cells under CART-19 therapy.
6. Our results indicate the necessity to extend the analysis to more isoforms and possibly to include the expression of splicing factors in screening approaches to identify patients at risk to of relapse under CART-19 therapy.
7. During alternative splicing, certain exons can be either included or excluded ("skipped"), thus leading to different transcript isoforms.

POINT-BY-POINT RESPONSES

We would like to thank the Reviewers for their time and the overall positive comments. Please find our point-by-point responses below. The respective changes in the manuscript are highlighted in yellow.

REVIEWER COMMENTS

Reviewer #1 (Remarks to the Author): with expertise in CD19 CAR-T

In this manuscript, Cortés-López et al. exploit a splicing reporter assay and a CD19 minigene to uncover (in an unbiased fashion) mutations in CD19 gene (from exon1 to exon3) that give rise to various CD19 isoforms. Using a mathematical approach, they assigned DNA mutations to protein isoforms. This study also suggests that some mutations can alter binding sites for RNA binding proteins (RBP) involved in mRNA splicing. Following an in silico screen, 11 RBP candidates were shortlisted and knocked-down in NALM-6 cell line. Finally, the authors uncovered that depletion of some RBP proteins and notably PTBP1 yields to non-functional CD19 isoform.

This work is well performed and contributes to the understanding of mechanisms leading to the alteration of CD19 expression, which is of significance to the field of CART-19 therapy. It also represents a helpful resource for CD19 mutations/isoforms.

I believe that this study will have more impact if the authors address the following points:

1) This manuscript is dedicated to CD19 mutations and their impact on CART-19 therapy, thus I expect the authors to go beyond an exhaustive list of mutations and provide a clearer answer as to how CD19 protein is lost in B-ALL:

- Figure 1 A & B. The authors observed an increase of Intron2-retention upon CART-19 therapy relapse. Before CART-19 therapy (so cells are supposed to be CD19+), Intron2-retention is clearly detected. This cannot result from the RNA contribution of rare CD19neg clones that pre-exist before CAR-T treatment, likely the aberrant isoform is already present in CD19+ B-ALL cells. What is the scenario proposed by the authors? Is there a monoallelic alteration in CD19+ B-ALL? Then a loss of heterozygosity upon treatment? Alternatively, do they envision a more efficient mis-splicing in CD19neg B-ALL (due to downregulation of a trans-acting factor such as PTBP1)?

We agree with the Reviewer that a clearer interpretation of our results will be beneficial.

To address the Reviewer's comments 1 and 2, we first assessed *CD19* splicing in immature and mature B-cells from healthy donors (PMID:30357359 and BLUEPRINT; PMID:27863955) (see also response to comment 2 below). In mature B-cells, *CD19* splicing is efficient, and thereby mostly fully spliced and functional *CD19* transcripts are generated (see **Figure I** below). In contrast, in the presumed CD19+ B-ALL before CART-19 therapy, we observe a broad range of intron2-retention levels, with 93% of patients exhibiting intron2-retention above 50%. We do not believe that increased intron2-retention in tumour cells is always caused by monoallelic alteration of *CD19*. Instead, the fact that tumour cells originate from incompletely differentiated B-cell precursors may explain increased intron2-retention, as immature B cells from healthy donors also show elevated levels of this isoform (see **Figure I** below). We think that the increased intron2-retention in immature and undifferentiated B-cells is mostly due to global changes in splicing factor expression (including *PTBP1*, *PCBP2* and *SF3B4*). This likely

leads to a partial downregulation of functional CD19 protein on the cell surface in most if not all B-ALL tumour cells.

Figure 1: CD19 alternative splicing in normal B cells (new Figure 1D)

For the development of complete CART-19 therapy resistance, we propose that due to tumour heterogeneity, some CD19+ B-ALL patients already host subclonal CD19neg B-ALL cells. These are selected under CART-19 therapy and will eventually lead to therapy resistance. That such subclonal CD19neg cells exist, has already been discussed in Rabilloud et al., 2021 (PMID:33558546).

We think that the cause of CD19 loss in subclonal cell lines can be manifold. Some will host mutations in *CD19* that further increase intron2-retention or lead to the production of other non-functional cryptic splice isoforms. Other subclonal CD19neg B-ALL cells could have genetically or epigenetically caused changes in splicing factor expression, leading to a further increase in intron2-retention (PTBP1, PCBP2 and SF3B4) or exon 2 skipping (SRSF3). Moreover, total *CD19* gene expression may be downregulated due to hypermethylation at the *CD19* locus as described in Ledererova et al., 2021 (PMID:34413165). In some cases, the complete loss of CD19 may involve the coincidence of several of these molecular alterations.

To present these hypotheses more clearly, we briefly mention the major observations in the revised Results section and discuss them in more detail in the revised Discussion:

“Likewise, alterations in the expression of *trans*-acting RBPs can induce aberrant *CD19* splicing, explaining the emergence of CD19-negative relapses in samples without mutations or with a low-allelic-frequency mutations or without mutations in the *CD19* locus. Interestingly, we find that the differentiation status of B-cells affects *CD19* splicing: in mature B-cells, almost complete exon 2 inclusion occurs, implying that all *CD19* transcripts give rise to functional CD19 protein. In contrast, intron 2 retention occurs in approximately half of the *CD19* transcripts in undifferentiated B-cell precursors (**Figure 1D**). Likewise, retention of intron 2 is predominant in B-ALL patient samples from the TARGET B-ALL cohort, with 93% of patients exhibiting retention frequencies above 50% (**Figure S1B**). Hence, incomplete B-cell differentiation in B-ALL may induce a transcriptional and posttranscriptional program, likely involving altered RBP expression, that reduces (but does not completely eliminate) the functional CD19 protein pool. This partial intron 2 retention predisposes the cancer cells to therapy resistance before they are actually subjected to CART-19 treatment, as observed in sorted B cell populations from a B-ALL patients before and after CART-19 therapy relapse [17]. For the development of complete CART-19 resistance, some B-ALL patients thus likely host

subclonal CD19-negative B-ALL cells which are further selected under the treatment [17]. The causes of complete CD19 loss in these subclonal cell populations are likely to be manifold, involving (epi)genetic changes such as hypermethylation of the *CD19* promoter [39], mutations in the *CD19* gene, splicing factor expression and combinations thereof.”

- Fig 1B. One B-ALL sample displays ~100% Intron2-retention frequency at both ‘initial screening’ and upon CART-19 relapse. What was the expression level of CD19 protein in this B-ALL sample before treatment? (I guess the B-ALL was initially CD19+, how this can be consistent with a 100% Intron2-retention frequency?)

We thank the Reviewer for spotting this discrepancy. We visually inspected the presumed RNA-seq datasets from patient #17 and realised that they are in fact DNA-seq samples that had been mislabelled in the Sequence Read Archive (SRA). This becomes evident when comparing the sequence coverage of these tracks to the RNA-seq and DNA-seq coverage tracks from other patients in this study (see **Figure II** below): The presumed RNA-seq tracks of patient #17 look very different from the other RNA-seq tracks and show no evidence of splicing at exon-intron boundaries. Instead, they clearly resemble the coverage tracks of the other DNA-seq samples.

In the revised version, we removed patient #17 from our analysis, as indicated in the Methods section, and updated **Figure 1B** accordingly.

Figure II: The two RNA coverage tracks from patient 17 (track 1&2) resemble DNA coverage (track 7-10) rather than RNA coverage (tracks 3-6).

-Strikingly, the manuscript starts (fig 1 A & B) and finishes (fig 5H & I) with the same ‘idea’: Intron2-retention is largely present in B-ALL samples. Fig 5H & I show that 93% of B-ALL possess more than 50% of Intron2-retention junction. With such numbers, why aren’t CD19neg B-ALL more prevalent (even without anti-CD19 therapeutic pressure)?

We agree that we missed some explanation on this topic in the manuscript. Before treatment, B-ALL is typically CD19+, since the initial remission rates for patients under CART-19 therapy

appear to be up to 90% (Davila & Brentjens, 2016, PMID:27930631). We do not think that this is contradicting our observation of high *CD19* intron2-retention. Arguably, intron2-retention likely gives rise to nonsense-mediated mRNA decay and thus to a decrease in CD19 at the protein level. However, this downregulation will not be complete, as a 50% and 75% share of intron2-retention it is expected to downregulate CD19 protein two- and four-fold, respectively. Hence, the cancer cells will still be CD19+ and, given the high efficiency of CARTs, will still be vulnerable to treatment.

However, high levels of intron2-retention likely predispose to the development of CART-19 resistance, since additional perturbations of *CD19* splicing are more likely to completely eliminate the CD19 surface protein pool in this background. Moreover, depending on the heterogeneity within the B-ALL cell population, some cells of a patient could have very high intron2-retention levels, while others have almost none. Consistent with such a scenario, a recent single-cell analysis of one patient before and after CART-19 therapy (Rabilloud et al., 2021, PMID:33558546), showed that some CD19neg B-ALL cells had already been present before CART-19 treatment. In conclusion, we suggest that increased intron2-retention in B-ALL leads to an overall CD19+ cell population, but promotes the development of CART-19 resistance by further splicing perturbations and/or strong heterogeneity between cells.

To address this issue in the manuscript, we now write in the revised Discussion that partial intron retention reduces, but does not completely eliminate CD19 protein production:

“(…) Hence, incomplete B-cell differentiation in B-ALL may induce a transcriptional and posttranscriptional program, likely involving altered RBP expression, that reduces (but does not completely eliminate) the functional CD19 protein pool. This partial intron 2 retention predisposes the cancer cells to therapy resistance before they are actually subjected to CART-19 treatment, as observed in sorted B cell populations from a B-ALL patients before and after CART-19 therapy relapse [17]. For the development of complete CART-19 resistance, some B-ALL patients thus likely host subclonal CD19-negative B-ALL cells which are further selected under the treatment [17]. The causes of complete CD19 loss in these subclonal cell populations are likely to be manifold, involving (epi)genetic changes such as hypermethylation of the *CD19* promoter [39], mutations in the *CD19* gene, splicing factor expression and combinations thereof.”

2) Is Intron2-retention detected in non-tumoral B cells?

To address this comment, we quantified intron2-retention levels in different immature and mature B-cell datasets (PMID:30357359 and BLUEPRINT; PMID:27863955). Interestingly, we observe high levels of intron2-retention (>50%) in undifferentiated B-cells but none in naive B-cells (see Figure I above). This could indicate a regulatory role of intron2-retention in B cell differentiation.

We included these results in the **new Figure 1D** and added the corresponding text to the revised Results (Section ‘CART-19 patients show increased *CD19* intron 2 retention after relapse’) and Discussion.

3) Are the mutations found at the relapse (figure S1) detected before CAR-T therapy?

The described mutations in Orlando et al. shown in Figure 1 were detected after CART-19 therapy. However, this does not exclude that they existed at low frequency already before the treatment, but there is currently no DNA sequencing data available at a stage before CART-19 therapy. We clarify this in the revised manuscript.

4) The authors indicate that the knockdown of PTBP1 (and in lesser extent PCBP2 and SF3B4) in NALM-6 cell line favours Intron2-retention and they showed in figure S5, that genes corresponding to these proteins are expressed in B-ALL. They should also show the expression of these genes in relapsed B-ALL. Especially, they should compare the expression level of PTBP1 in relapsed B-ALL samples harbouring Intron2-retention versus samples before treatment.

In this context, the authors could also complement figure 5H and show the expression of PTBP1 (and other RBP genes of interest) in the B-ALL TARGET cohort.

We thank the Reviewer for this suggestion and agree that it is important to assess whether *PTBP1* expression is modulated in B-ALL before vs. after relapse from CART-19 therapy.

As a starting point, we performed the corresponding analyses in the Orlando dataset which provides pre-processed expression data for nine patients before and after CART-19 relapse. We found that *PTBP1* expression showed inconsistent changes upon relapse: In five out of nine patients, *PTBP1* tended to increase upon relapse compared to the levels at screening, whereas it decreased in the other four patients. We did, however, detect a consistent and significant upregulation of *PTBP2* mRNA expression (also known as *nPTB*), which is tightly repressed by the PTBP1 protein via alternative splicing and nonsense-mediated mRNA decay (Spellman et al., 2007, PMID:17679092) and hence serves as a direct sensor for PTBP1 activity in the cells. Thus, the observed increase of *PTBP2* mRNA expression may indicate a reduced PTBP1 protein function in the B-ALL cells after CART-19 therapy relapse.

Next, we moved to the TARGET B-ALL data and analysed the RBP expression patterns as suggested (see also response to comment 7 of Reviewer #2). Briefly, we observed that B-ALL tumours, which typically show higher intron2-retention compared to normal B-cells, on average also showed reduced levels of *PTBP1* expression as expected from our working hypothesis (see **Figure III (A)** below). Furthermore, within the TARGET B-ALL cohort, intron2-retention levels and *PTBP1* showed a significant inverse correlation (see **Figure III (B)** below), i.e., patients with high intron2-retention tend to be characterised by low *PTBP1* expression as expected based on our working hypothesis. Moreover, we observed a strong correlation between *CD19* intron2-retention and increased *PTBP2* mRNA levels, acting again as a sensor for lowered PTBP1 protein activity.

Figure III: PTBP1 and PTBP2 mRNA levels correlate with CD19 intron2-retention. (A) Scatterplots comparing mRNA levels to intron2-retention frequency for 220 B-ALL patient samples from TARGET B-ALL data. (B) PTBP1 mRNA levels are variable, but PTBP2 mRNA levels are significantly increased upon CART-19 therapy relapse in B-ALL patients from Orlando et al. (new Figure 6A, B)

The new analyses have been in the **new Figures 6A, B and S7A** and described in detail in the Results section ‘:

“In line with a role of PTBP1 in *CD19* mis-splicing in tumours, we find that patient samples from the TARGET B-ALL cohort on average show lower *PTBP1* mRNA expression compared to healthy B-cells (**Figure S7A**). Within the B-ALL samples, *PTBP1* expression negatively correlates with *CD19* intron2-retention, as expected based on our knockdown experiments ($R = 0.24$; **Figure 6A**, left). In addition, we investigated *PTBP2* mRNA expression, which is tightly repressed by the PTBP1 protein via alternative splicing and nonsense-mediated mRNA decay [31] and hence serves as a direct sensor for PTBP1 activity in the cells. Indeed, we find a strong correlation between increased *PTBP2* mRNA levels, i.e., lowered PTBP1 protein activity, and increased *CD19* intron2-retention ($R = 0.56$; **Figure 6A**, right). To test for changes upon CART-19 relapse, we extracted *PTBP1* and *PTBP2* from expression data provided by the Orlando study [5]. Although we do not detect systematic changes in the *PTBP1* mRNAs levels, the *PTBP2* mRNA levels are significantly increased at relapse relative to screening, possibly indicating lowered PTBP1 protein levels (P value = 0.037, Wilcoxon rank-sum test; **Figure 6B**). Together, these analyses suggest that PTBP1 is a regulator of *CD19* alternative splicing, which we decided to explore further.”

5) Concerning mathematical modelling, it seems that the model is less robust for the prediction of intron2-retention and alt-exon2 (fig 3B and S3B). Can the authors comment on that?

The Reviewer is correct that some isoforms are predicted better by the model than others. As shown in **Figure 3B**, the cross-validation is worse for intron2-retention and alt-exon2, with Pearson correlation coefficients of around 0.7 between model and data. In contrast, the other isoforms are predicted with correlation coefficients of ~0.9.

One possible explanation for this low prediction power is the low abundance of the isoforms, and therefore the lower signal-to-noise ratio compared to the most abundant isoforms (inclusion, skipping). In fact, in the WT minigene, intron2-retention, alt-exon2 and alt-exon3 exhibit isoform frequencies of <5% (**Figure 2C**). Since most of the minigenes in the screen show isoform patterns close to WT, the signal-to-noise ratio is worse for such lowly abundant isoforms.

Why is the prediction power better for alt-exon3? Most likely because of the more pronounced accumulation of alt-exon3 in response to mutations: In the mutant minigene population, intron2-retention and alt-exon2 mostly accumulate to an isoform frequency of <25%, whereas alt-exon3 rises to up to 90% and has a much larger proportion of minigenes >25% (**Figure 2C**). This stronger accumulation of alt-exon3 is also reflected in the magnitude of single-mutation effects predicted by the model (**Figure 3C**) The stronger mutation effects on alt-exon3 compared to the other low abundance isoforms likely improves the signal-to-noise ratio in the calculation of the Pearson correlation coefficient between model and data. Taken together, we reason that the lower prediction power for alt-exon2 and intron2-retention is due to lesser signal contained in the measurements of these isoforms.

We added a shortened version of these considerations to the revised Results section ‘Therapy-impacting isoforms accumulate in response to numerous point mutations’, in which we describe the regression model.

6) The authors uncovered that CD19 exon 1-3 minigene can generate several cryptic isoforms. Yet, they did not show whether some of those described isoforms really exist in B cells/B-ALL primary samples.

We screened TARGET B-ALL and normal B-cell data for the cryptic isoforms. Indeed, we found several patients and also normal B-cell samples in which two of the cryptic junctions (present in four of our predicted cryptic isoforms) occurred (see splice junction quantifications in **Figure 1C, D, S1A**). We did not find evidence for the other cryptic junctions. Since most of them appear only upon mutation in our screen, it is unlikely that they are abundant in normal B-cells or B-ALL samples. We therefore think that these isoforms will only accumulate under CART-19 selection, although they may already exist subclonally (see above).

Currently, directly testing for the accumulation of cryptic isoforms upon CART-19 treatment in patients is difficult, as only a handful of CART-19 relapse patient RNA-seq samples are available so far (Orlando et al., 2018, PMID:30275569). Furthermore, these samples are only available as mapped reads and cryptic isoforms might not be captured with standard mapping procedures. Hence, we believe that in-depth future analyses of patient material will tell to which extent cryptic isoforms contribute to CART-19 therapy resistance.

We now discuss the presence of cryptic isoforms in patients in the revised Results section ‘Cryptic isoforms destroy the *CD19* open reading frame and are associated with recurrent mutations’ and elaborate on the issue in slightly more detail in the revised Discussion. The description in Results reads:

“Screening for the occurrence of the 96 cryptic isoforms in the TARGET B-ALL patient samples, we readily detected two junctions of cryptic junctions that had been present already prior to CART-19 therapy (**Figure 1C, S1A**). Other cryptic isoforms predicted from our screen were not found in these patients that had not been treated with CART-19 therapy, but could already exist subclonally and/or may only emerge under selective pressures of CD19-directed immunotherapy. The same applies to the associated mutations identified from our screen which were also not present in the TARGET B-ALL data (**Table S5**).”

7) The authors should describe the PTBP1 iCLIP2 assay.

We apologise for missing a section on iCLIP2. We now include a description on the iCLIP2 approach in the Methods section ‘PTBP1 iCLIP2 experiments’.

8) DeepRiPe predicts changes of RBP binding upon DNA mutations. If I well understood only altered (“knocked-out”) mutations are considered. Have the authors tried to do the analysis the other way round: by defining which mutations result in a knock-in of a RPB binding site?

We apologise that this was unclear in the manuscript. Actually, we do consider both, binding sites removed or generated by a mutation. The direction of change is encoded in the sign of the delta score predicted by DeepRiPe (see **Table S7**, worksheet “DeepRiPe mutations”).

In the revised manuscript, we explicitly state in the Results and Methods sections as well as in the legend for **Table S7** that mutations were considered for both scenarios.

Reviewer #2 (Remarks to the Author): with expertise in alternative splicing, cancer

In this paper, Cortés-López et. al. study how aberrant alternative splicing of CD19 leads to loss of cognate CD19 epitope necessary for success of CART-19 immunotherapy for management of B-cell acute lymphoblastic leukemia. Focusing specifically on CD19 exons 1-3, the authors utilize a minigene system to conduct a high-throughput mutagenesis screen. In conjunction with quantitative modelling using multinomial logistic regression, the authors identified novel intronic and exonic cis-regulatory mutations that give rise to alternative isoforms that encode for non-functional CD19 proteins. Furthermore, through in silico analyses and knockdown of RNA-binding proteins (RBPs), the authors characterized trans-regulators that promote production of CART-19 sensitive isoforms. While there are sections where the connection between experimental results to patient data is strenuous, the authors, utilize rigorous bioinformatic approaches to offer insight on a novel mechanism of treatment resistance. Further experiments are needed to strengthen their conclusions.

Major comments:

1. Fig. 1. Patient mutations cause splicing changes in the CD19 minigene. Except for mutations from patients 4, 5, 14, and 15, it is not clear that the isoform frequencies caused by the mutations observed in other treatment refractory patients are significantly different from the wildtype control. Inclusion of statistical tests will be helpful here. Furthermore, do the splicing patterns observed in the minigene (Fig. 1F) match that of the patient data (Fig. 1B)?

We apologise for not indicating significance. We now calculated the significance of the minigene splicing changes induced by patient mutations (Student's *t*-test). Mutations from patients #2, #4, #5, #14 and #15 show significant changes.

Due to the high variability in the patient data, it is difficult to quantitatively compare isoform levels in patients to the minigene results. However, we find that the two patients with mutations that lead to a significant increase in intron2-retention in our minigene (patients #4 and #14) also show increased intron2-retention upon CART-19 relapse.

In the revised version of the manuscript, we added information on significant changes in the former Figure 1F (now **Figure 1I**). We also labelled patients #4 and #14 in **Figure 1B** and describe the link as follows:

“(…) while mutations from patients #4 and #14 (#14.2) cause intron 2 retention. The latter mirrored the increase of this isoform in the same patients after CART-19 therapy relapse (**Figure 1B**).”

2. Fig. 2. The accuracy of the screen can be strengthened by a subset of RT-PCR validations.

We agree with the Reviewer's remark that RT-PCR validation experiments can further support the accuracy of our screen. The ultimate goal of our screen is to predict single mutation effects on *CD19* splicing using a regression model. Thus, we analysed minigenes containing single point mutations using RT-PCR. We chose 19 single mutations in regions with significant effects on at least one of the isoforms according to the regression prediction (cf. **Figure 3C**): these regions included the 3' and 5' ends of exon 2, the 3' end of exon 3 and an internal region of exon 2 (around position 650). *CD19* minigenes harbouring the chosen mutations (see **Figure IV** below) were generated by site-directed mutagenesis, transfected into HEK293 cells and splicing outcomes were determined by RT-PCR and quantitative capillary gel electrophoresis.

Figure IV: Individual point mutations tested by RT-PCR (new Figure 3E)

As predicted by our screen, mutations in exon 2 gave rise to increased skipping and alt-exon2 isoforms, whereas mutations in exon 3 yielded enhanced intron2-retention and alt-exon3 (Figure IV above). In quantitative terms, the overall correlation between regression predictions and the RT-PCR measurements resulted in a high correlation coefficient of $R=0.84$ (Figure V below). At the individual isoform level, we obtained the following correlation coefficients between the predictions and RT-PCR analysis: inclusion, $R=0.73$; skipping, $R=0.74$; intron2-retention, $R=0.94$; alt-exon2, $R=0.82$; alt-exon3, $R=0.88$ (see new Figure S3D in the revised manuscript). Taken together, the direct RT-PCR validation quantitatively confirms the single mutation predictions of our regression model.

Figure V: Correlation of RT-PCR validations with the screening results (new Figure 3F)

We included these results in the revised manuscript by adding three new figures (Figure 3E, F, S3D) which are described in the Results section ‘Therapy-related isoforms accumulate in response to numerous point mutations’. Furthermore, we updated the Methods accordingly.

3. Fig. 3. In the cross-validation, is the 10% that is left out never seen by any of the validation process? Or is a different 10% left out in each validation? This clarification is important as in the latter case, the validation result is influenced by the modeling from the same cohort and is not considered as a true validation.

We thank the Reviewer for this comment and agree that clarification will be helpful. In Figure 3B, we performed a 10-fold cross-validation and compared the correlation between model and data for each of the five isoforms. As far as we understand, the Reviewer is concerned that during the prediction of an isoform, the model might have been trained based

on data from the very same minigenes (using measurements of the other isoforms). We agree that this would be problematic. However, in our approach, the left-out 10% of the minigenes were never seen by any validation process and the softmax regression model was trained using all five isoforms from the remaining 90%. Then, we independently predicted all five isoforms for the left-out 10% of minigenes, implying that the validation data was never seen during training.

To exclude that biases exist in the splitting in calibration and validation data, this procedure was repeated 10 times using random splits between training (90%) and test (10%) data ('10-fold cross-validation'). Before the cross-validation, the dataset was once randomly divided into 10 equally sized parts, and then each of the 10 sections was used once as test data (i.e., each data point is only 1x part of the test data). Using this approach, we saw little variability in the prediction power for these ten validation runs and therefore report the average correlation coefficient in **Figure 3B**.

We clarified the description of the cross-validation approach in the revised Methods (see 'Estimation of single mutation effects and splicing-affecting mutations').

4. Fig. 4. Mutations generated in the high-throughput mutagenesis screen gives rise to cryptic isoforms. Are any of the mutations from the high-throughput assay overlap with known mutations in B-ALL patients?

We now screened mutation data from B-ALL patients (TARGET B-ALL programme) subjected to chemotherapy, but not CART-19 therapy. In these samples, we could not find mutations overlapping with mutations inducing cryptic isoforms in our screen. We think that most of such mutations will be subclonal and are only detectable upon selection during CART-19 therapy. As mentioned above, only a small number of CART-19 relapse patients have been characterised by sequencing so far. Therefore, we envision that the availability of more patient samples in the future will tell to which extent mutations causing cryptic isoforms contribute to CART-19 therapy resistance.

In the revised manuscript, we included statements in the Results section 'Cryptic isoforms destroy the *CD19* ORF and are associated with recurrent mutations' in which we clarify that mutations associated with cryptic isoforms are not found in the TARGET B-ALL cohort:

"Screening for the occurrence of the 96 cryptic isoforms in the TARGET B-ALL patient samples, we readily detected two junctions of cryptic junctions that had been present already prior to CART-19 therapy (**Figure 1C, S1A**). Other cryptic isoforms predicted from our screen were not found in these patients that had not been treated with CART-19 therapy, but could already exist subclonally and/or may only emerge under selective pressures of CD19-directed immunotherapy. The same applies to the associated mutations identified from our screen which were also not present in the TARGET B-ALL data (**Table S5**)."

In the Discussion, we write:

"To assess the role of the predicted cryptic splice isoforms in patients, we screened sequencing data from the TARGET B-ALL cohort and indeed recurrently found two junctions from the cryptic isoforms that we had observed in the mutagenesis data. Even though other cryptic junctions were absent and mutations associated with cryptic isoforms according to screen were also not found in the patient data, these may still

emerge during CART-19 selection. Currently, there is a shortage of large-scale sequencing data of patient material before and after CART-19 therapy [53]. Future analysis of such data with a special focus on cryptic splice site usage will be important to identify mutations or splice isoforms that are predictive for CART-19 therapy success.”

5. Fig. 4C. CRISPR replacement of the mutant sites should be performed to indicate that mutations cause the change of splicing and loss of CD19 protein production on cell surface. This will be needed for the conclusion that high-throughput mutagenesis identifies mutations controlling CART-19 therapy resistance.

We agree with the Reviewer that this will be an interesting experiment. However, in line with Reviewer #3, we believe that such an analysis is beyond the scope of the current paper. For the present work, we therefore focussed on the role of *trans*-acting factors in controlling CD19 surface protein levels (see following comment) and think that CRISPR replacement will be an avenue for a follow-up study.

We included the following sentence in the revised Discussion:

“In the future, targeted CRISPR/Cas9 replacement experiments using the endogenous *CD19* gene should be performed to validate that the predicted mutations cause physiological changes in splicing, loss of CD19 protein exposure on cell surface and CART-19 therapy resistance.”

6. Fig. 5. Do PTBP1 KD cells show decreased CD19 protein on cell surface? This information is needed to connect PTBP1’s regulation on CD19 splicing and CART-19 therapeutic resistance. Moreover, it is not clear to the reviewer whether CD19 intron retention causes CART-19 therapy resistance. Are the functional domains deleted when the intron is retained? Were there reports to show causality of CD19 intron retention in therapy resistance?

We agree with the Reviewer that it is crucial to show that the PTBP1-mediated changes in *CD19* alternative splicing affect CD19 protein surface exposure. To test this, we performed siRNA-mediated knockdown of *PTBP1* in P493-6 and MHHCALL4 cells, two human B cell lines derived from B cell lymphoma and pre B-ALL, respectively, and analysed CD19 surface exposure using flow cytometry. Importantly, we found that depletion of *PTBP1* leads to reduced levels of CD19 protein on the surface of both cell lines (see **Figure VI** below). This suggests that the PTBP1-mediated splicing changes could contribute to CART-19 therapy resistance.

Figure VI: CD19 surface exposure upon PTBP1 knockdown (new Figure 6D-F)

These new and important results are shown in **Figure 6D-F** and **Figure S8** and described in the following paragraph from the Results (see 'Depletion of PTBP1 and several other RBPs results in non-functional CD19 isoforms'):

"Next, we chose to assess whether PTBP1-mediated splicing changes affect CD19 surface exposure on B-cells. To test this, we performed siRNA-mediated knockdown of *PTBP1* in P493-6 and MHHCALL4 cells (**Figure S8A, B**) and confirmed that the knockdown increased levels of *CD19* intron2-retention in both cell lines (**Figure S8C, D**). Then, we measured CD19 protein surface expression using CD19 antibody staining and flow cytometry analysis. Strikingly, we found that both cell lines show reduced CD19 surface exposure upon *PTBP1* depletion (**Figure 6D-F and S8E**). Thus, by interfering with CD19 protein expression on the cell surface, *PTBP1* depletion could indeed contribute to CART-19 therapy resistance."

As highlighted in **Figure 2B**, intron2-retention introduces a premature termination codon that likely reduces *CD19* transcript and protein expression via nonsense-mediated mRNA decay. Regarding its potential impact on CART-19 therapy, we initially suggested that intron2-retention could contribute to CART-19 therapy resistance (Asnani et al., 2019, PMID:31591467) as referred to in our Introduction. This was further supported by a single-cell RNA-seq study of CD19pos and CD19neg B cell populations from a B-ALL patient who underwent relapse after CART-19 therapy: Targeted splicing quantifications on the B cell populations confirmed that the CD19neg cells after CART-19 therapy relapse almost completely retained intron 2, "explaining the absence of CD19 protein despite the presence of *CD19* mRNA." (Rabilloud et al., 2021, PMID:33558546).

To emphasise this, we mention the putative association of intron2-retention with nonsense-mediated mRNA decay and reduced protein expression in the Results section:

"PTBP1 stands out among the putative regulators as it shows the strongest effects on intron2-retention. This splicing event introduces a premature termination codon that likely reduces *CD19* transcript and protein expression via nonsense-mediated mRNA decay (**Figure 2B**)."

We also refer to the reported enrichment of intron2-retention in CD19neg B-ALL cells in the Discussion:

"Moreover, in a comparison of B cells from a B-ALL patient, it was found that intron 2 retention had already occurred prior to CART-19 therapy (CD19-positive B cells) and had become predominant in the CD19-negative B cells after relapse [17]."

7. Fig. 5 H. The authors analyzed RNA-seq data of 220 B-ALL patients. Are there CD19 mutations matching the known mutations or the high-throughput results? What are the levels of PTBP1? Do they show correlations with the levels of CD19 intron retention? A significant correlation would serve as a separate validation supporting the premise of the study. An absence of correlation could suggest that other mechanisms rather than mutations and PTBP1 cause CD19 intron retention.

To address the Reviewer's question, we first analysed *CD19* mutations in the TARGET B-ALL cohort. In brief, we utilised somatic mutations that had been identified in whole-exome DNA-sequencing data for TARGET B-ALL patients using the NCI Genomic Data Commons (GDC) DNA-seq Analysis Pipeline. In total, we could identify 39 patients with somatic mutations within

CD19. These included 11 mutations in the region corresponding to our minigene (exons 1-3), neither of which showed an effect on *CD19* splicing in our high-throughput screen. This result, however, needs to be treated with caution, since the datasets apparently comprise a mixture of whole-genome and whole-exome sequencing and are only available in a pre-processed format, making it hard to judge whether mutations were identified comprehensively. Moreover, it is important to keep in mind that these patients had not been treated with CART-19 therapy. As for the cryptic isoforms, we hypothesise that splicing-effective mutations could already exist subclonally in the B-ALL cells and/or may emerge under therapy selection.

In the revised version, we included the 11 mutations that were identified in TARGET B-ALL in the new worksheet “All annotated variants” in **Table S5**.

We further analysed the levels of *PTBP1* mRNA expression as suggested. Compared to normal B-cells, the B-ALL samples, on average, show reduced *PTBP1* expression (see **Figure VII** below) which may higher levels of intron2-retention in these samples.

Figure VII: RBP expression in normal B-cells and B-ALL patient samples (new **Figure S7A**)

Within the cohort of B-ALL samples, both *PTBP1* expression and intron2-retention show pronounced variation. As suggested by the Reviewer, we related both species by correlation analysis (see below) and found that high *PTBP1* expression was indeed significantly associated with lower intron2-retention ($R = -0.24$; see **Figure III (A)** in response to Reviewer #1). These observations further support our hypothesis that *PTBP1* negatively regulates *CD19* intron 2 retention.

In addition, we investigated *PTBP2* (also known as *nPTB*) expression which is tightly regulated by *PTBP1* via a mechanism involving alternative splicing and nonsense-mediated mRNA decay (Spellman et al., PMID:17679092). Due to this cross-regulation mechanism, low levels of *PTBP1* result in an upregulation of *PTBP2* expression. Hence, *PTBP2* levels can serve as an indicator of *PTBP1* protein levels. Comparing *PTBP2* levels to intron2-retention, we observe a significant positive correlation ($R = 0.56$; see **Figure III (A)** in response to Reviewer #1). This supports our hypothesis that low levels of *PTBP1* protein (indicated by increased levels of *PTBP2* mRNA) result in increased *CD19* intron 2 retention.

These results were included in the revised manuscript by adding the **new Figures S7A and 6A**. The results are described in the Results section ‘Depletion of *PTBP1* and several other RBPs results in non-functional *CD19* isoforms’.

8. The observation in Fig. 5H that all B-ALL patients show CD19 retained intron from 30% to nearly 100% suggest that CD19 intron retention is commonly observed in B-ALL. Does CD19 show retained intron in normal B cells? Results in Fig. 5H imply that the level of intron retention determines therapy resistance and not its presence since all patients showed intron retention. Is this supported by clinical data?

The Reviewer's question about intron2-retention in normal B-cells was addressed in our response to Reviewer #1 (comment 2). Briefly, we discuss there that the levels of intron2-retention are negligible in mature B-cells, whereas the isoform accumulates in undifferentiated B-cell precursors and in B-ALL. The corresponding results are shown in the **new Figure 1D** and presented in the Results section 'Depletion of PTBP1 and several other RBPs results in non-functional CD19 isoforms' of the revised manuscript.

Regarding the second point, there are currently too little RNA sequencing datasets after CART-19 relapse available to address this question. To avoid confusion: more RNA-seq samples exist after B-ALL relapse from chemotherapy, but these datasets do not allow us to assess whether intron2-retention levels play a causal role in CART-19 therapy resistance. However, an important point we can make is that intron2-retention significantly increases upon CART-19 therapy resistance as shown in **Figure 1B**.

In the revised discussion, we mention the shortage of genome-wide transcriptome data after CART-19 relapse and point out that such analyses will be an interesting avenue of research for assessing the role of alternative splicing in CART-19 resistance.

Minor comments:

1. Fig. 1. The authors state that the minigene generates the same isoforms as the endogenous CD19 gene. However, in both gel-like representation and quantification of semi-quantitative RT-PCR, there appears to be higher quantity of the intron2-retention isoform in the minigene construct. Inclusion of p-value for the quantification would be beneficial.

The Reviewer correctly points out that the relative abundance of the intron2-retention isoform is slightly elevated in the minigene-derived transcripts compared to endogenous *CD19* splicing. Minor deviations of a minigene reporter are not uncommon, since it represents a minimal regulatory system which can recapitulate many but not all regulatory relationships that may act on the endogenous splicing event. The aim of the former Figure 1D, E (now **Figure 1F, G**) was therefore to show that the minigene gives rise to the same isoforms as the endogenous *CD19* gene, and that these occur at least at similar levels.

In the revised version, we rephrase the respective sentence in the results section to present this aspect more clearly. In addition, following the Reviewer's suggestion, we performed a one-way ANOVA test to show that the isoform frequencies do not significantly differ between the *CD19* WT minigene and the endogenous *CD19* gene in NALM-6 cells. The information was added to the legend of the figure (now **Figure 1G**).

2. Fig. 1D. The difference in splicing pattern observed in K562 cells can be cell type specific. It cannot be compared with NALM-6. The reviewer recommends elimination of K562 results.

The splicing measurements from K562 cells were included as an additional control to show that the *CD19* WT minigene gives rise to similar isoforms and were not intended to show any differences between the two cell lines. Given that the ENCODE data used for RBP binding

site prediction (via DeepRiPe) were in large parts generated in K562 cells, we think that it is useful as a marginal note that the *CD19* splicing pattern is preserved to a certain extent in this cell line. We therefore decided to keep the measurements in the former Figure 1D (now **Figure 1F**).

Reviewer #3 (Remarks to the Author): with expertise in alternative splicing, RPB

The publication by Cortes-Lopez, Schulz, Enculescu et al. describe a deep dive into the regulatory landscape of CD19 exon 2 processing, including detailed mutational analysis and mathematical modeling to characterize the landscape of complex splicing mis-regulation that can occur with this event. I think this is both quite interesting not only biologically (as mis-splicing of CD19 exon 2 is important physiologically, as well-described in the text), but also as a great model for how such detailed analysis of an individual event can reveal unexpected complexity in splicing regulation (and how different experimental and modeling approaches can be used to understand the mechanisms behind this complexity).

The only significant issues I would raise are:

1) Since they are based off of actual publicly available CLIP datasets, I find it somewhat peculiar to only use the abstracted 'predicted binding sites' for analysis in Figure 5; although I think the analyses presented are well-described, it seems to me to be fairly obvious (and easy) to also calculate and show these overlaps using the actual ENCODE & PAR-CLIP datasets (particularly for the RBPs shown in Fig. 5D, and particularly as many of them (TAF15, SF3B4, PUM2, PTBP1, HNRNPM, HNRNPK, and FUS, based on a quick scan of the ENCODE website) have data in K562, which seems like it could be reasonably similar to the B-ALL sample type under study here).

We agree with the Reviewer that it would be advantageous to show actual measurements of RBP binding, and not only predicted binding sites. The vast majority of eCLIP experiments by the ENCODE consortium were performed in the two cell lines K562 and HepG2. However, *CD19* is a B-cell-specific marker and even though K562 cells are from a hematopoietic origin (bone marrow lymphoblasts of a CML patient), they are from a myeloid lineage and show hardly any *CD19* expression (see last column in **Figure I** above) nor do the HepG2 cells which were derived from a hepatocellular carcinoma (not shown).

Thus, due to the low *CD19* expression in the ENCODE cell lines, the eCLIP coverage is very sparse for *CD19* and only a few reads have been captured for the investigated RBPs. To illustrate this, we show below the eCLIP coverage tracks for TAF15, SF3B4, PUM2, PTBP1, HNRNPM, HNRNPK, and FUS, among others, in HepG2 and K562 cells (see **Figure VIII** below). The isolated reads do not pile up such that reliable peak calling and identification of significant and reproducible binding sites would be possible.

To circumvent this issue, we used the recently published prediction algorithm DeepRiPe (Ghanbari & Ohler, 2020, PMID:31992613) that had been trained on the transcriptome-wide ENCODE eCLIP and allows to infer the impact of point mutations on RBP binding. This enabled us to still predict RBP binding sites in the *CD19* pre-mRNA sequence.

In the revised manuscript, we added this information to present this aspect more clearly:

“Furthermore, we included RBP binding information from the public resource of ENCODE eCLIP datasets. Since the *CD19* mRNA is hardly expressed in the ENCODE cell lines and binding events in *CD19* can therefore not be directly extracted, we employed the prediction algorithm DeepRiPe [29].”

Figure VIII: Only few individual reads appear within CD19 in the eCLIP datasets from HepG2 and K562 cells. In the region spanning exon 1-3, there is only a single read.

2) I think the conclusions PTBP1 section of Fig. 5 should either be written less strongly, or requires additional data and analyses (I'd recommend the former). In particular, to me the DeepRiPe sites in Fig. 5G shows little overlap with the strongest region of iCLIP2 crosslink events, and the ATtRACT sites while overlapping that region also show a large number throughout the entire event (raising questions as to whether they have such a high false positive rate with the PTBP1 motif to be uninformative), so I don't know that I agree that the data presented supports the conclusion "The broad binding at splicing-effective positions and beyond supports that PTBP1 is a <<direct and central regulator>> of CD19 alternative splicing, with most prominent effects on intron 2 retention." (emphasis added by me)

We agree with the Reviewer and have now toned down our conclusions as suggested. The cited sentence now reads:

"This suggests that PTBP1 directly regulates CD19 splicing via intron 2 binding."

3) It would also be helpful throughout to add some additional background rates for comparison throughout. For example, line 284-286 describe the fraction of mutations in close proximity to (real or cryptic) splice sites – what is the background rate for all nucleotides in this region and how enriched is this?

We fully agree and now analysed background rates of random mutations being close to a splice site as suggested by the Reviewer. In the revised manuscript, we now report odds ratios and *P* values from Fisher's exact test at this and several further positions where we report an enrichment:

Line 308-311: “We found that the majority of mutations with a prevalence score > 0.25 are either in close proximity or directly overlap with the associated cryptic splice site (77.4% with distance < 5 nt; odds ratio 7.55, P value = $1.793e-07$, Fisher’s exact test; **Figure 4E**).”

Line 249-251: “Inspecting in more detail the 83 mutations that specifically impact on *CD19* exon 2 skipping, we find them to cluster within and around exon 2 (odds ratio 2.06 for such mutations to occur inside exon 2, P value = 0.002614, Fisher’s exact test).”

Line 427-429: “Moreover, DeepRiPe predicts 78 mutations in 63 positions that change PTBP1 binding, out of which 10 are splicing-affecting in our screen (odds ratio 3.21, P value = 0.002481, Fisher’s exact test).”

4) I would recommend being more explicit with some conclusions – e.g. ‘Taken together, our results strongly suggest that *CD19* mutations contribute to CART-19 therapy resistance by inducing splicing changes’ – unless I’m mistaken, the contribution to CART-19 therapy resistance is entirely inferred in this paper (based on predictions of whether the splicing change observed would create either a loss of exon 2 or frameshift, and based on the assumption that the impact of the mutation observed in the minigene reporter will be recapitulated in the full transcript). It would be more correct to say something like ‘our results indicate that far more *CD19* mutations are predicted to create isoforms that would escape CART-19 recognition’

We agree with the Reviewer and carefully revised this and further sentences to be more precise. The respective sentence now reads:

“Thus, our results indicate that far more *CD19* mutations can create isoforms that would escape CART-19 recognition.”

Minor comments:

- I’m confused by the term ‘(near-)constitutive exons like *CD19* exon 2’ (line 422) – from the author’s data (and a quick skim of K562 and GM12878 data on the UCSC browser), I wouldn’t refer to this as near-constitutive (as intron 2 retention in particular seems relatively common)

We agree with the Reviewer that this was misleading. We now write:

“For *CD19*, we find that strong mutation effects are mainly centred around canonical and cryptic splice sites, whereas in other examples such as *MST1R* exon 11 or *FAS* exon 6, mutation effects are more dispersed across intronic and exonic sequences [19,38]. This suggests that *CD19* exon 2 splicing may be controlled by multiple splicing enhancers that act redundantly and render inclusion less sensitive to individual point mutations [20]. Therefore, *CD19* exon 2 may require more specific perturbations and as we show here, does not only respond with exon skipping, but tends to employ alternative splice sites and intron retention, both of which are clinically relevant in the case of CART-19 therapy resistance.”

Reviewer #4 (Remarks to the Author): with expertise in alternative splicing, bioinformatics

“High-throughput mutagenesis identifies mutations and RNA-binding proteins controlling CD19 splicing and CART-19 therapy resistance” by Mariela Cortés-López et al., provides a detailed investigation of the CD19 (and exons 1-3 in particular) using high-throughput mutagenesis, mathematical modelling and RBP knockdowns to understand the nucleotides, cis-elements and binding sites that regulate splicing, and the complex isoforms generated by mutations in these elements.

The manuscript provides two contributions. Firstly, the study provides a detailed dissection of the impact of mutations on splicing which is often complex and difficult to predict. By performing a high-throughput mutagenesis screen the authors are able to dissect the contribution of nucleotides to splicing and their impact on the resulting gene architecture. This analysis showed that splice mutations can markedly impact isoform diversity, particularly at exons with weakly competitive splice sites. Accordingly, this provides a systems-level understanding of the splicing code and the impact of mutations on splicing diversity (which is remarkably large).

The second strength of the study is the clinical relevance of the CD19 gene analysed. Mutations to CD19 drive resistance in BALL patients to CAR-T cell therapy. Understanding the role of splicing mutations to this gene may ultimately identify mechanisms to prevent resistance and, more immediately, the data may provide a resource for the interpretation of mutations in BALL patients and provide prognostic markers of CART-19 therapy resistance.

More broadly, the study is well designed, and the manuscript is well-written, with clear figures, rigorous analysis, and fair interpretation of results. The methods are detailed, with data and script appropriately available. I congratulate the authors on the study.

Major points.

1. I do not have any major concerns with the study designs, analysis or interpretation. However, one suggestion on how the study may be improved would be to provide a greater context for the splicing and expression of the CD19 gene in healthy and BALL patient populations. This would largely involve an analysis of CD19 gene splicing in publicly available RNA-seq data from healthy RNAseq datasets (such as GTex) and from B-ALL patients from the Therapeutically Applicable Research To Generate Effective Treatments (TARGET) program. Given the authors have identified a diversity of splicing junctions using long-read data, these would form useful annotations against which to analyse publicly available short-read data for alternative isoforms that have been otherwise missed.

I realise that this has been performed to varying degrees in some previous studies, and this may be why the authors have not specifically focused on this analysis (indeed the authors present some of this data in the manuscript (such as Figure S6D), however, I believe that foregrounding this analysis would provide the readers with an understanding of the CD19 landscape, and provides a useful context in which to consider the suitability of the CD19 mini-gene assay. This includes how well the CD19 recapitulates healthy and patient splicing (quantitatively and splicing structure)?, as well as interpret the outcomes, including the impact of mutations and resulting alternative isoforms. Attempting to recapitulate the impact of these mutations and their complex splicing outcomes in vivo (using gene editing etc.) would be ideal, however, I realise that this is a large undertaking and outside the scope of this current study.

In line with the Reviewer’s suggestion (and with comments of Reviewer #1), we now provide a more in-depth analysis of *CD19* exon 1-3 splicing in B-cells from healthy donors and B-ALL patients. Briefly, we show that exon 2 inclusion is the dominant isoform in healthy B-cells (near 100%; see **Figure I** above). In undifferentiated B-cell precursors, *CD19* mRNA is also expressed, but the splicing pattern is altered, as exon 2 inclusion and intron2-retention each contribute around 50% (see **Figure I** above). Likewise, intron2-retention contributes 30%-100% in B-ALL samples (former Figure 5H, now **Figure 1C**), and this pattern is not different between primary tumours and those relapsed from chemotherapeutic treatment. Hence, incomplete differentiation of B-cells may generally promote intron2-retention and thereby *CD19* protein downregulation. The sum of exon 2 skipping and cryptic isoforms is generally low (<20%) in both healthy B-cell precursors and B-ALL samples, suggesting that these isoforms may accumulate only in a subpopulation of tumour cells or upon selection by CART-19 treatment.

There is no large-scale RNA sequencing data available before and after CART-19 therapy relapse besides the nine paired patient samples from Orlando et al. that we already analysed in our study. Therefore, we could not comprehensively assess the role of mutations and downstream splicing changes in CART-19 resistance. However, as the Reviewer states, our dataset is a useful resource for future analyses to identify relevant mutations and splice junctions once more CART-19 therapy relapse datasets become available.

Please refer to our responses to Reviewer #1 for a more detailed description. We added the respective information to the revised Results and Discussion sections and included the splicing quantifications in B-cells from healthy donors in the **new Figure 1D**.

In line with the Reviewer, we think that *in vivo* gene editing will be an interesting avenue for future studies and briefly discuss this as an outlook in the revised Discussion.

2. One notable difference is the BALL patients from Orlando study often harbor more complex mutations (deletions or insertions greater than 5nt in length, Table S2), whilst my understanding is that the error-prone PCR genetics smaller single-point mutations. It would be helpful to provide a comparison of the mutations (type and quantity) for (i) Orlando study, (ii) Orlando study (iii) within healthy populations (gnomAD) and (iv) within BALL and cancer patient populations (COSMIC, ClinVar etc.).

Following the Reviewer’s suggestion, we compared the frequency of SNVs and more complex mutations from the different resources. In total, we collected 830 mutations in the *CD19* region corresponding to our minigene (chr16:28931871-28933144), including 24 indels longer than 5 nt. The frequencies of these indels in the different datasets are summarised in **Table I** below:

	ClinVar	COSMIC	Ensembl	gnomAD	Orlando et al.	TARGET
Total	42	113	505	189	22	11
SNVs	41 (97.6%)	110 (97.3%)	478 (94.7%)	179 (94.7%)	2 (9.1%)	11 (100%)
Indels (>1 nt)	1 (2.4%)	3 (2.7%)	26 (5.1%)	10 (5.3%)	20 (90.9%)	0 (0%)
Indels (>5 nt)	1 (2.4%)	1 (0.9%)	12 (2.4%)	4 (2.1%)	8 (36.4%)	0 (0%)

Table I: Frequencies of complex indels (>5 nt length) in the different datasets

The frequency of indels seems to be largely comparable between datasets, with the notable exception of the data from the Orlando study which, as pointed out by the Reviewer, are dominated by large indels. However, since the original DNA sequencing data for these patients are not accessible and limited information is available about their processing, it is difficult to judge to what extent this observation reflects true biological enrichment versus technical biases. Moreover, beyond Orlando et al., a handful of studies in patients with B-cell related diseases have reported individual mutations in the context of CART-19 therapy, comprising mostly short indels and SNVs (e.g., PMID:33023981, 16672701, 17882224, 20445561, 32881995), but these numbers are hardly representative. We therefore conclude that with the limited data available at present, it is not possible to draw reliable conclusions about the distribution of mutation types affecting *CD19* after CART-19 therapy.

In order to make these data accessible to readers, we added an additional worksheet to **Table S5**, summarising all the 830 SNVs which we collected for the *CD19* minigene region.

3. The authors show the SpliceAI predictions correlate relatively well with the outcomes from the high-throughput mutagenesis study and suggest their data can be used to benchmark tools for splicing detection. However, this correlation also suggests that there may be value in performing a more detailed analysis of SpliceAI predictions across the broader *CD19* gene (beyond exon 2). Whilst these predictions aren't as rigorous as the mutagenesis assay, they could nevertheless provide a broader landscape in which to interpret mutations that impact *CD19* splicing that may drive CAR-T cell resistance. For example, and 'predicted' set of spliceAI elements across the *CD19* gene could be similarly analysed with respect to publicly available RNA-seq data and mutation databases (from healthy and BALL patients).

In line with the Reviewer's suggestion, we generated SpliceAI predictions across the complete *CD19* gene. **Figure IX** below shows the maximum SpliceAI score per position, separated for the gain or loss of a given splice site. The colouring further indicates an overlap with variants reported in publicly available databases (gnomAD, ClinVar, COSMIC V94, Ensembl and TARGET). In total, SpliceAI predicts 285 mutations with a score > 0.2 ("high recall"), including 37 that overlap with reported variants.

Figure IX: SpliceAI predictions along the complete *CD19* gene (new Figure S5C)

The results of this analysis have been added as the **new Figure S5C**. In addition, we provide the full list of SpliceAI predictions for the *CD19* gene and their overlap with annotated variants in the **new Table S6**. The results are described in the main text as follows:

“Due to the robust performance of SpliceAI, we decided to predict splice-changing mutations throughout the entire *CD19* gene and overlapped them with publicly reported single nucleotide variants (SNVs; **Figure S5C**). These predictions and variant overlap are provided as a resource (**Table S6**) and can be used to evaluate the impact of new patient mutations on *CD19* splicing in the future.”

Minor points. The manuscript is very well written, and the figures are clear. I have only a few suggested minor grammatical revisions;

We rephrased the following sentences in the revised manuscript.

1. Several studies reported that in 40-60% of cases the cancerous B-cells become invisible to the CARTs due to loss of detectable CD19 epitope (CD19-negative).

Now reads “Several studies reported that in 40-60% of relapse cases, the cancerous B-cells become invisible to the CARTs because they lose expression of the CD19 epitope (CD19-negative) [5-8].”

2. Taken together, our dataset is a comprehensive resource for prognostic markers of CART-19 therapy resistance and for a systems-level understanding of the splicing code.

Now reads “Taken together, our dataset allows for a systems-level understanding of the splicing code and provides a comprehensive resource of prognostic markers for CART-19 therapy resistance.”

3. Altogether, the *in silico* predictions suggest the presence of an extensive RBP network controlling CD19 splicing that may impact on the CART-19 therapy success.

Now reads “Overall, the *in silico* predictions suggest the presence of an extensive RBP network that controls *CD19* splicing and may impact the CART-19 therapy success.”

4. Moreover, an upregulation of PTBP1 has been implicated in the acquired resistance of pancreatic ductal carcinoma cells to the chemotherapeutic drug gemcitabine.

Now reads “In addition, upregulation of PTBP1 has been implicated in acquired resistance to the chemotherapeutic agent gemcitabine in pancreatic ductal carcinoma cells [46].”

5. Thus, besides the regulation of protein expression, other factors like cellular availability may further impact on PTBP1 function in B-ALL cells under CART-19 therapy.

Now reads “Thus, in addition to regulation of PTBP1 expression, other factors such as availability may also influence PTBP1-mediated regulation in B-ALL cells under CART-19 therapy.”

6. Our results indicate the necessity to extend the analysis to more isoforms and possibly to include the expression of splicing factors in screening approaches to identify patients at risk to of relapse under CART-19 therapy.

Now reads “Our results point to the need to extend the analysis to additional *CD19* isoforms and to incorporate the expression of splicing factors in screening approaches to identify patients at risk of relapse on CART-19 therapy.”

7. During alternative splicing, certain exons can be either included or excluded (“skipped”), thus leading to different transcript isoforms.

Now reads “In alternative splicing, certain exons can be either included or excluded (“skipped”), resulting in different transcript isoforms.”

REVIEWERS' COMMENTS

Reviewer #1 (Remarks to the Author):

In this revised manuscript Cortes-Lopez et al. have addressed my concerns and clarified several points. In particular, they showed that intron2-retention is not restricted to B-ALL: it also occurs in healthy developing/immature B-cells (but not in naïve B cells). The proposed scenario is that this CD19 mis-splicing is induced by a decrease of PTBP1 activity. However, downregulation of PTBP1 expression in two B-ALL cohorts (Orlando dataset and TARGET B-ALL) is not as clear-cut as expected. To prove their point the authors used PTBP2 expression as a surrogate marker of PTBP1 protein activity. They showed that PTBP2 is more expressed in B-ALL, suggesting that PTBP1 protein is less active. Finally, their PTBP1 knock-down assays in cell lines show that PTBP1 down-regulation induces a decrease of CD19 protein expression.

Overall, the new data presented strengthen their conclusions.

Minor point:

Indeed, Rabilloud et al (ref #17) showed that Intron2-retention exists prior CAR-T cell therapy, yet they did not analyse exon 2 skipping. Thus, I suggest to indicate only ref #16 in line 107: 'exon 2 skipping were observed to pre-exist in patients prior to CART-19 therapy [16,17]'; as well as in line 534: 'The pre-existence of isoforms skipping exon 2 or exons 5-6 has been previously discussed as a possible biomarker [16,17].'

Reviewer #2 (Remarks to the Author):

The authors addressed the comments well. I do not have additional comments and recommend for publication.

Reviewer #3 (Remarks to the Author):

I appreciate the authors' detailed revisions and comments, and they have addressed my major issues. As noted before, I think this is quite interesting work that will be highly useful for the community

One minor comment for the revised manuscript - the minigene analysis added in Fig 3E I think is interesting, but I'd appreciate some notation for where those mutations are relative to the splice sites – in particular, I think (based on Fig. 3D) that the A474 mutations are mutating the splice site 'AG' itself, which while a nice control is (to me) quite different than other mutations further in the intron or exon in terms of how impressive or interesting it is to see such significant splicing changes. (Semi-related to this, the reference to Table S5 on line 262 I think is a bit weirdly phrased, as on reading it I expected it to have information on the 19 point mutations used in Fig. 3E)

Reviewer #4 (Remarks to the Author):

Dear Authors, Thank you for your considered responses to my questions that have been addressed, and for incorporating my feedback into your manuscript. Congratulations on a well designed, written and informative study that provides functional insight into the impact of mutations on the CD19 gene expression and splicing more broadly.

REVIEWERS' COMMENTS

We would like to thank the Reviewers for the very constructive reviewing process.

Reviewer #1 (Remarks to the Author):

In this revised manuscript Cortes-Lopez et al. have addressed my concerns and clarified several points. In particular, they showed that intron2-retention is not restricted to B-ALL: it also occurs in healthy developing/immature B-cells (but not in naïve B cells). The proposed scenario is that this CD19 mis-splicing is induced by a decrease of PTBP1 activity. However, downregulation of PTBP1 expression in two B-ALL cohorts (Orlando dataset and TARGET B-ALL) is not as clear-cut as expected. To prove their point the authors used PTBP2 expression as a surrogate marker of PTBP1 protein activity. They showed that PTBP2 is more expressed in B-ALL, suggesting that PTBP1 protein is less active. Finally, their PTBP1 knock-down assays in cell lines show that PTBP1 down-regulation induces a decrease of CD19 protein expression.

Overall, the new data presented strengthen their conclusions.

Minor point:

Indeed, Rabilloud et al (ref #17) showed that Intron2-retention exists prior CAR-T cell therapy, yet they did not analyse exon 2 skipping. Thus, I suggest to indicate only ref #16 in line 107: 'exon 2 skipping were observed to pre-exist in patients prior to CART-19 therapy [16,17]'; as well as in line 534: 'The pre-existence of isoforms skipping exon 2 or exons 5-6 has been previously discussed as a possible biomarker [16,17].'

We thank the Reviewer for spotting this. We removed reference #17 from the sentence as suggested.

Reviewer #2 (Remarks to the Author):

The authors addressed the comments well. I do not have additional comments and recommend for publication.

Reviewer #3 (Remarks to the Author):

I appreciate the authors' detailed revisions and comments, and they have addressed my major issues. As noted before, I think this is quite interesting work that will be highly useful for the community

One minor comment for the revised manuscript - the minigene analysis added in Fig 3E I think is interesting, but I'd appreciate some notation for where those mutations are relative to the splice sites – in particular, I think (based on Fig. 3D) that the A474 mutations are mutating the splice site 'AG' itself, which while a nice control is (to me) quite different than other mutations further in the intron or exon in terms of how impressive or interesting it is to see such significant splicing changes. (Semi-related to this, the reference to Table S5 on line

262 I think is a bit weirdly phrased, as on reading it I expected it to have information on the 19 point mutations used in Fig. 3E)

To allow for an easy overview of the tested mutations, a schematic showing the position of the different mutations within the minigene and relative to the splice sites has been included in the new **Supplementary Figure 5a**.

We have added a new worksheet to the supplementary table (now Supplementary Data 3) which provides details on the 19 point mutations that were tested in **Figure 3e**. In addition, all measurements are listed in detail in the new file **Source Data 2**, worksheet "Fig. 3e, S5b,c".

Reviewer #4 (Remarks to the Author):

Dear Authors, Thank you for your considered responses to my questions that have been addressed, and for incorporating my feedback into your manuscript. Congratulations on a well designed, written and informative study that provides functional insight into the impact of mutations on the CD19 gene expression and splicing more broadly.